# Offline Goal-conditioned Reinforcement Learning with Quasimetric Representations

**Vivek Myers**    **Bill Chunyuan Zheng**    **Benjamin Eysenbach**[†]    **Sergey Levine**

UC Berkeley    [†]Princeton University

## Abstract

Approaches for goal-conditioned reinforcement learning (GCRL) often use learned state representations to extract goal-reaching policies. Two frameworks for representation structure have yielded particularly effective GCRL algorithms: (1) *contrastive representations*, in which methods learn "successor features" with a contrastive objective that performs inference over future outcomes, and (2) *temporal distances*, which link the (quasimetric) distance in representation space to the transit time from states to goals. We propose an approach that unifies these two frameworks, using the structure of a quasimetric representation space (triangle inequality) with the right additional constraints to learn successor representations that enable optimal goal-reaching. Unlike past work, our approach is able to exploit a **quasimetric** distance parameterization to learn **optimal** goal-reaching distances, even with **suboptimal** data and in **stochastic** environments. This gives us the best of both worlds: we retain the stability and long-horizon capabilities of Monte Carlo contrastive RL methods, while getting the free stitching capabilities of quasimetric network parameterizations. On existing offline GCRL benchmarks, our representation learning objective improves performance on stitching tasks where methods based on contrastive learning struggle, and on noisy, high-dimensional environments where methods based on quasimetric networks struggle.

## 1 Introduction

Learning temporal distances lies at the heart of many important problems in both control theory and reinforcement learning. In control theory, such distances form important Lyapunov functions [1] and control barrier functions [2], and are at the core of reachability analysis [3] and safety filtering [4] In Reinforcement Learning (RL), such distances are important not just for safe RL [5], but also for forming value functions in tasks ranging from navigation [6] to combinatorial reasoning [7] to robotic manipulation [8, 9]. Ideally, these learned distances have two important properties: *(i)* they can encode paths that are shorter than those demonstrated in the data (i.e., stitching); and *(ii)* they can capture long-horizon distances with low variance.

Current methods for learning temporal distances typically only achieve one of these properties. Methods based on Q-learning [10, 11, 12] stitch trajectories with Temporal Difference (TD) updates to find shortest paths, but often produce compounding errors that make it challenging to apply to long-horizon tasks [13]. Monte Carlo methods [14, 15] can directly learn goal-reaching value functions, which can be connected to temporal distances [16], but their ability to find *shortest* paths remains limited. Methods based on learning a quasimetric geometry [17, 18, 16], which impose a triangle inequality over distances as an architectural invariance, do find shortest paths and don't require dynamic programming with compounding errors, but fail in stochastic settings and/or when learning from off-policy (suboptimal) data.

Website and code: `https://tmd-website.github.io/`

39th Conference on Neural Information Processing Systems (NeurIPS 2025).

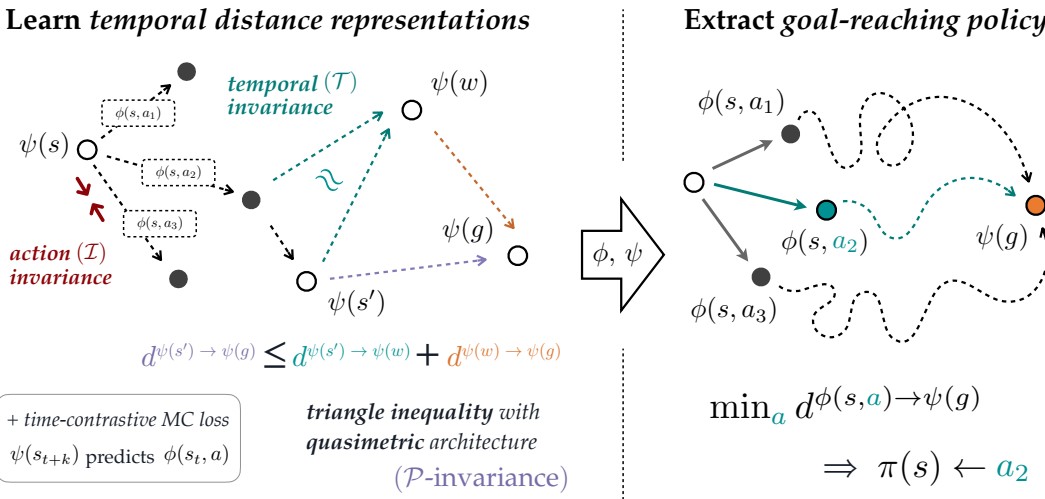

Figure 1: *(Left)* TMD learns a temporal distance $d_\theta$ that satisfies the triangle inequality and action invariance. It does this by minimizing the distance between the learned distance and the distance between the successor features of the states and actions in the dataset. *(Right)* The learned distance is used to extract a goal-conditioned policy.

The aim of this paper is to build a method for learning temporal distances that retains the long-horizon estimation capabilities of Monte Carlo methods but nonetheless is able to compute shortest paths. We take an invariance perspective to do this. Temporal distances satisfy various invariance properties. Because they are value functions, they satisfy the Bellman equations. Prior work has also shown that they satisfy the triangle inequality, even in stochastic settings [18, 16]. The triangle inequality, also a form of invariance [19], is powerful because it lets us architecturally winnow down the hypothesis space of temporal distances by only considering neural network architectures that satisfy the triangle inequality [17, 20].

Importantly, the fact that temporal distances satisfy the triangle inequality holds for *any* temporal distance, including both optimal temporal distances and those learned by Monte Carlo methods. This raises an important question: might there be an *additional* invariance that is satisfied by optimal temporal distances, but not those learned by Monte Carlo methods? Identifying such invariance properties that would enable us to use Monte Carlo methods to architecturally winnow the hypothesis space, and then use this additional invariance property to identify optimal temporal distances within that space.

Our key contribution is a method for learning optimal goal-reaching distances that combines the long-horizon, probabilistic inference of Monte Carlo temporal distances with the optimality and stitching capabilities of quasimetric architectures. We use a quasimetric architecture that imposes the triangle inequality as an architectural constraint, combined with two additional invariance properties that apply at the *transition* level. When these invariances are enforced as constraints on features learned with Monte Carlo estimation, they impose a structure roughly similar to the Bellman optimality equations across the space of goal-conditioned (Kroenecker Delta) reward functions.

We translate these invariance properties into a practical method for Goal-Conditioned Reinforcement Learning (GCRL) that we call Temporal Metric Distillation (TMD). To the best of our knowledge, TMD is the first GCRL method that uses a quasimetric value parameterization to implicitly stitch behaviors, while also learning optimal policies in stochastic settings with suboptimal data. On benchmark tasks of up to 21-dimensions as well as visual observations, we demonstrate that our method achieves results that considerably outperforms that of similar baselines. Additional experiments reveal the importance of the enforced invariances and contrastive learning objective. Given the importance of long-horizon reasoning in many potential applications of RL today, we believe our work is useful for thinking about how to learn optimal temporal distances.

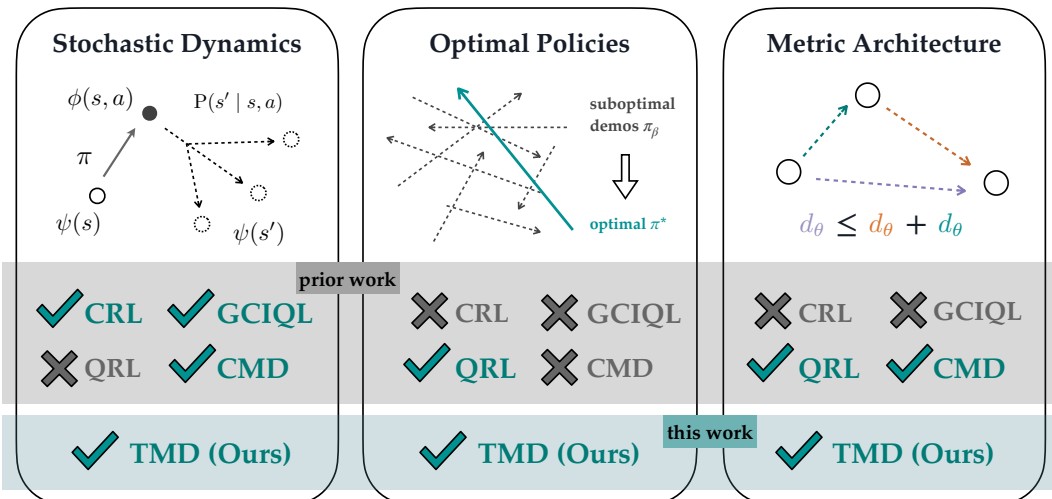

Figure 2: TMD enables key capabilities over prior work: *(Left)* handling stochastic transition dynamics, *(Center)* learning optimal policies from offline data, and *(Right)* stitching behaviors as a property of network architecture.

## 2 Related Work

Our method provides a unifying framework connecting temporal distance learning to (optimal) offline GCRL. The resulting method gets the benefits of both, learning optimal goal-reaching policies from offline data with stochastic dynamics, and using a quasimetric architecture to optimally stitch together behaviors without the compounding errors of TD learning.

### 2.1 Temporal Distances

We build on prior approaches to learning *temporal* distances, which reflect the reachability of states [16]. Temporal distances are usually defined as the expected number of time steps to transit from one state to another [21, 22]. Recent work has provided probabilistic definitions that are also compatible with continuous state spaces and stochastic transition dynamics [16]. A key consideration when thinking about temporal distance is *which* policy they reflect: is this an estimate of the number of time steps under our current policy or under the optimal policy? We will use *optimal temporal distance* to mean the temporal distance under the optimal (distance-minimizing) policy.

Algorithmically, this choice is often reflected in the algorithm one uses for learning temporal distances. Methods based on Q-learning typically estimate optimal temporal distances [11, 23, 24], and are often structurally similar to popular actor-critic methods. Some quasimetric methods also learn optimal temporal distances in deterministic MDPs by enforcing the triangle inequality as an architectural constraint, effectively computing shortest paths in a directed graph [17, 18]. Prior work has shown that temporal distance learning can be important for finding paths that are better than those demonstrated in the data, and can enable significantly more data efficient learning [25] (akin to standard results in the theory of Q-learning [26]).

Methods based on Monte Carlo learning typically operate by sampling pairs of states that occur nearby in time (though not necessarily temporally-adjacent); distances are minimized for such positive pairs, and maximize for pairs of states that appear on different trajectories [27, 15]. These Monte Carlo methods often estimate the temporal distance corresponding to the policy that collected the data. Methods for goal-conditioned behavioral cloning [28, 29], though not directly estimating temporal distances, are effectively working with this same behavioral temporal distance [7]. Despite the fact that Monte Carlo methods do not estimate optimal temporal distances, they often outperform their Q-learning counterparts, suggesting that it is at least unclear whether the errors from learning the behavioral (rather than optimal) temporal distance are larger or smaller than those introduced by TD learning's compounding errors. Our work bridges these two notions of temporal distance,

providing a method that learns optimal temporal distances while reducing the reliance on TD learning to propagate values (and accumulate errors).

## 2.2 Offline Reinforcement Learning

Our investigation into temporal distances closely mirrors discussions in the offline RL literature about 2-step RL methods [30], which often use Monte Carlo value estimation, versus multi-step RL methods [31], which often use Q-learning value estimation. These 1-step RL methods avoid the compounding errors of Q-learning, yet are limited by their capacity to learn $Q_g^*(s, a)$ rather than $Q^\beta$. However, their strong performance over the years [32, 15] suggests it is an open question whether the compounding errors of Q-learning outweigh the benefits of learning the behavioral value function, rather than the value function of the optimal policy.

## 3 Temporal Metric Distillation (TMD)

In this section, we formally define TMD in terms of the invariances it must enforce to recover optimal distances, and by extension, the optimal policy. In Section 4 we will then show how these invariances can be converted into losses which can be optimized with a quasimetric architecture that enforces the triangle inequality.

### 3.1 Notation

We consider a controlled Markov process $\mathbf{M}$ with state space $\mathcal{S}$, action space $\mathcal{A}$, and dynamics $\mathrm{P}(s' \mid s, a)$. The agent interacts with the environment by selecting actions according to a policy $\pi(a \mid s)$, i.e., a mapping from $\mathcal{S}$ to distributions over $\mathcal{A}$. We further assume the state and action spaces are compact.

Policies $\pi \in \Pi$ are defined as distributions $\pi(a \mid s)$ for $s \in \mathcal{S}, a \in \mathcal{A}$. When applicable, for a fixed policy $\pi$, we can denote the state and action at step $t$ as random variables $\mathfrak{s}_t$ and $\mathfrak{a}_t$, respectively. We will also use the shorthand

$$\mathfrak{s}_t^+ \triangleq \mathfrak{s}_{t+K} \text{ for } K \sim \mathrm{Geom}(1 - \gamma). \tag{1}$$

We equip $\mathbf{M}$ with an additional notion of *distances* between states. At the most basic level, a distance $\mathcal{S} \times \mathcal{S} \to \mathbb{R}$ must be non-negative and equal zero only when passed two identical states. We will denote the set of all distances as $\mathcal{D}$, defined as

$$\mathcal{D} \triangleq \{d : \mathcal{S} \times \mathcal{S} \to \mathbb{R} : d(s, s) = 0, d(s, s') \geq 0 \text{ for each } s, s' \in \mathcal{S}\}.$$

A desirable property for distances to satisfy is the triangle inequality, which states that the distance between two states is no greater than the sum of the distances between the states and a waypoint [18]. A distance satisfying this property is known as a *quasimetric*. Formally, we construct

$$\mathcal{Q} \triangleq \{d \in \mathcal{D} : d(s, g) \leq d(s, w) + d(w, g) \text{ for all } s, g, w \in \mathcal{S}\}. \tag{2}$$

If we further restrict distances to be symmetric ($d(x, y) = d(y, x)$), we obtain the set of traditional metrics over $\mathcal{S}$.

### 3.2 TMD Operators

TMD learns a distance parameterization that is made to satisfy two constraints: (i) the *triangle inequality*,

$$d(x, z) \leq d(x, y) + d(y, z) \text{ for any } x, y, z \in \mathcal{S} \times \mathcal{A} \cup \mathcal{S}, \tag{3}$$

and (ii) *action invariance*,

$$d\big(s, (s, a)\big) = 0 \text{ for any } s \in \mathcal{S} \text{ and } a \in \mathcal{A}. \tag{4}$$

We will show that to ensure that we recover the optimal distance $d_{SD}$ [16] given the learned (backward NCE) contrastive critic distance, the missing additional constraint is a form of consistency over the environment dynamics with respect to the expected (exponentiated) distances. This constraint

resembles the "SARSA"-style Bellman consistency, which backs up values by averaging over dynamics to learn on-policy values. So, what TMD is doing with these additional constraints is weakening the form of Bellman consistency that is required to recover the optimal distance from the standard $\max_{a \in \mathcal{A}}$ Bellman operator to the weaker on-policy SARSA Bellman operator. TMD thus turns on-policy SARSA into an off-policy algorithm through the metric constraints.

We can define this additional constraint as the fixed point of the following operator:

$$\mathcal{T}(d)(x,y) = \begin{cases} -\log \mathbb{E}_{P(s'|s,a)}\left[e^{-d(s',y)}\right] - \log \gamma & \text{if } x = (s,a) \in \mathcal{S} \times \mathcal{A}, \\ d(x,y) & \text{otherwise.} \end{cases} \tag{5}$$

The triangle inequality Eq. (3) and action invariance Eq. (4) properties can also be written in terms of operator fixed points:

$$\mathcal{P}(d)(x,z) \triangleq \min_{y \in \mathcal{S}} \left[ d(x,y) + d(y,z) \right] \tag{6}$$

$$\mathcal{I}d(s,x) \triangleq \begin{cases} 0 & \text{if } x = (s,a) \\ \mathcal{I}(d)(s,x) & \text{otherwise.} \end{cases} \tag{7}$$

### 3.3 Properties of path relaxation

Path relaxation $\mathcal{P}$ [19] (Eq. 6) enforces invariance to the triangle inequality, i.e., $\mathcal{P}(d) = d$ if and only if $d \in \mathcal{Q}$.

**Theorem 1.** *Take $d \in \mathcal{D}$ and consider the sequence*

$$d_n = \mathcal{P}^n(d).$$

*Then, $d_n$ converges uniformly to a fixed point $d_\infty \in \mathcal{Q}$.*

In light of Theorem 1 we denote by $\mathcal{P}_* = \lim_{n \to \infty} \mathcal{P}^n$ the fixed point operator of $\mathcal{P}$, and note that $\mathcal{P}_*$ is in fact a projection operator onto $\mathcal{Q}$.

Proofs of Lemmas 5 to 7 and Theorem 1 can be found in Appendix C.

### 3.4 The modified successor distance

The *modified successor distance* $d_{\mathrm{SD}}^\pi \in D$ can be defined by [16]: 11.5

$$d_{\mathrm{SD}}^\pi(x,y) \triangleq \begin{cases} 0 & \text{if } x = y, \\ -\log p^\pi\left(\frac{P(\mathfrak{s}^+ = g | \mathfrak{s}_0 = s, \mathfrak{a}_0 = a)}{P(\mathfrak{s}^+ = g | \mathfrak{s}_0 = g)}\right) \text{ for } K \sim \gamma & \text{if } x = (s,a) \in \mathcal{S} \times \mathcal{A}, y = g \in \mathcal{S} \\ -\log \mathbb{E}_{\pi(a|s)}\left[e^{-d_{\mathrm{SD}}^\pi((s,a),g)}\right] - \log \gamma & \text{if } x = s \in \mathcal{S}, x \neq y \\ d_{\mathrm{SD}}^\pi(s,g) - \log \pi(a \mid g) & \text{if } y = (g,a) \in \mathcal{S} \times \mathcal{A}. \end{cases} \tag{8}$$

The *optimal successor distance* $d_{\mathrm{SD}}^*$ can then be stated as

$$d_{\mathrm{SD}}^*(x,y) \triangleq \min_{\pi \in \Pi} d_{\mathrm{SD}}^\pi(x,y). \tag{9}$$

This distance is useful since it lets us recover optimal goal-reaching policies. For any $s, g \in \mathcal{S}, a \in \mathcal{A}$, the distance is proportional to the optimal goal-reaching value function

$$d_{\mathrm{SD}}^*((s,a),g) \propto_a -Q_g^*(s,a) \tag{10}$$

where $Q_g^*(s,a)$ is defined as the standard optimal $Q$-function for reaching goal $g$ [15]:

$$Q_g^*(s,a) \triangleq \max_{\pi \in \Pi} \mathbb{E}_{\{\mathfrak{s}_i, \mathfrak{a}_i\} \sim \pi} \left[ \sum_{t=0}^{\infty} \gamma^t \, P(\mathfrak{s}_t = g \mid \mathfrak{s}_0 = s, \mathfrak{a}_0 = a) \right]. \tag{11}$$

and

$$V_g^*(s) \triangleq \max_{a \in \mathcal{A}} Q_g^*(s,a). \tag{12}$$

In fact, we can equivalently define $d_{\mathrm{SD}}^*\big((s,a),g\big)$ in terms of $Q^*$:

$$d_{\mathrm{SD}}^*\big((s,a),g\big) = \log V_g^*(g) - \log Q_g^*(s,a). \tag{13}$$

Similar to Myers et al. [16], we argue that contrastive learning can recover these distances, i.e.,

$$\mathcal{C}(\pi) = d_{\mathrm{SD}} \tag{14}$$

Then, through the operators in Section 3.2, we will extend this to the optimal distance $d_{\mathrm{SD}}^*$.

For convenience, we also define the set of realized successor distances

$$\widetilde{\mathcal{D}} \triangleq \{d_{\mathrm{SD}}^\pi : \pi \in \Pi\}. \tag{15}$$

Note that $\widetilde{\mathcal{D}}$ does not necessarily contain the optimal distance $d_{\mathrm{SD}}^*$, as no single policy is generally optimal for reaching all goals.

**Remark 2.** *The optimal successor distance $d_{\mathrm{SD}}^*$ satisfies*

$$d_{\mathrm{SD}}(s,(s,a)) = 0 \text{ for all } s \in \mathcal{S} \text{ and } a \in \mathcal{A}.$$

## 3.5 Convergence to the optimal successor distance

Applying the invariances in Section 3.2 to the contrastive distance Eq. (14), the TMD algorithm can be defined symbolically as

$$\mathcal{M}(\pi) \triangleq (\mathcal{P}_* \circ \mathcal{T} \circ \mathcal{I})^\infty \mathcal{C}(\pi). \tag{16}$$

In other words, TMD computes the initial $\pi_\beta$ distance $\mathcal{C}(\pi)$, and then enforces the invariance (architecturally or explicitly), as expressed with the iterative application of $\mathcal{T} \circ \mathcal{I}$ followed by projection onto $\mathcal{Q}$ by $\mathcal{P}_*$.

**Theorem 3.** *The TMD algorithm converges pointwise to the optimal successor distance $d_{\mathrm{SD}}^*$ for any policy $\pi$ with full state and action coverage, i.e.,*

$$\lim_{n \to \infty} (\mathcal{P}_* \circ \mathcal{T} \circ \mathcal{I})^n \mathcal{C}(\pi) = d_{\mathrm{SD}}^*. \tag{17}$$

Our approach for proving Theorem 3 will be to analyze the convergence properties of $(\mathcal{P}_* \circ \mathcal{T} \circ \mathcal{I})$ over the space of "suboptimal" distances $\mathcal{D}_+^*$, defined as

$$\mathcal{D}_+^* \triangleq \{d \in \mathcal{D} : d(x,y) \geq d_{\mathrm{SD}}^*(x,y) \text{ for all } x,y \in \mathcal{S} \times \mathcal{A} \cup \mathcal{S}\}. \tag{18}$$

Unfortunately, $(\mathcal{P}_* \circ \mathcal{T} \circ \mathcal{I})$ is not a contraction on $\mathcal{D}_+^*$, so we cannot directly apply the Banach fixed-point theorem as we would for the standard Bellman (optimality) operator. Instead, we will show this operator induces a "more aggressive" form of tightening over $\mathcal{D}_+^*$, which will allow us to prove convergence to $d_{\mathrm{SD}}^*$. We start by showing that $d_{\mathrm{SD}}^*$ is a fixed point of $(\mathcal{P}_* \circ \mathcal{T} \circ \mathcal{I})$ in Lemma 4.

**Lemma 4.** *The optimal successor distance $d_{\mathrm{SD}}^*$ is the unique fixed point of $\mathcal{P}_* \circ \mathcal{T} \circ \mathcal{I}$ on $\mathcal{D}_+^*$.*

Proofs are in Appendix B.

# 4 Implementing TMD

We show that the backward NCE contrastive learning algorithm can recover an initial estimate of $d_{\mathrm{SD}}^{\pi_\beta}$. As justified by Theorem 3, we can then enforce the invariances to recover the optimal distance $d_{\mathrm{SD}}^*$.

The algorithm learns a distance $d_\theta$ parameterized by a quasimetric neural network $\theta$ such as Metric Residual Network (MRN) [17]. By construction, this distance is a quasimetric that is invariant to $\mathcal{P}$, i.e., $\mathcal{P}d_\theta = d_\theta$.

## 4.1 Initializing the Distance with Contrastive Learning

Defining the critic

$$f(s, a, g) \triangleq -d_\theta\big((s, a), g\big),$$

the core contrastive objective is the backward NCE loss:

$$\mathcal{L}_{\text{NCE}}\big(\phi, \psi; \{s_i, a_i, s'_i, g_i\}_{i=1}^N\big) = \sum_{i=1}^N \log\left(\frac{e^{f(s_i, a_i, g_i)}}{\sum_{j=1}^N e^{f(s_j, a_j, g_i)}}\right) \tag{19}$$

which is enforced across batches of triplets $\{s_i, a_i, s_{i+k}\}_{i=1}^N$ for $k \sim \text{Geom}(1 - \gamma)$ sampled from the dataset generated by policy $\pi_\beta$.

The optimal solution to this objective is

$$f(s, a, g) = \log\left(\frac{\text{P}(\mathfrak{s}^+ = g \mid \mathfrak{s} = s, \mathfrak{a} = a)}{\text{P}(\mathfrak{s}^+ = g)C(g)}\right). \tag{20}$$

for some $C(g)$ [33].

The parameterization $f(s, a, g) = -d_\theta\big((s, a), g\big)$ where $d_\theta$ is a quasimetric-enforcing parameterization (see [17, 18]) ensures that

$$C(g) = \frac{\text{P}(\mathfrak{s}^+ = g \mid \mathfrak{s} = g)}{\text{P}(\mathfrak{s}^+ = g)},$$

so the only valid quasimetric satisfying Eq. (20) is $d_\theta = d_{\text{SD}}{}^{\pi_\beta}$.

Optimality of $\mathcal{L}$ in Eq. (19) implies that the learned distance $d_\theta = \mathcal{C}(\pi_\beta) = d_{\text{SD}}{}^{\pi_\beta}$.

The additional invariance constraints $\mathcal{I}$ and $\mathcal{T}$ can be directly enforced by regressing $\|d_\theta - \mathcal{I}d_\theta\|_\infty$ and $\|d_\theta - \mathcal{T}d_\theta\|_\infty$ to zero. Theorem 3 guarantees that if we can enforce those constraints and enforce invariance to $\mathcal{P}$ by using a quasimetric architecture (e.g., MRN [17]), we can recover the optimal distance $d_{\text{SD}}^*$.

In practice, we will directly enforce the constraints across the batches used in our contrastive loss. We will use the MRN parameterization for $d_\theta$ for $\theta = (\psi, \phi)$ on learned representations of states ($\psi$) and state-action pairs ($\phi$):

$$d_\theta\big(s, g\big) \triangleq d_{\text{MRN}}(\psi(s), \psi(g)) \qquad\qquad d_\theta\big((s, a), g\big) \triangleq d_{\text{MRN}}(\phi(s, a), \psi(g))$$
$$d_\theta\big(s, (s, a)\big) \triangleq d_{\text{MRN}}(\psi(s), \phi(s, a)) \qquad d_\theta\big((s, a), (s', a')\big) \triangleq d_{\text{MRN}}(\phi(s, a), \phi(s', a'))$$

where

$$d_{\text{MRN}}(x, y) \triangleq \frac{1}{K} \sum_{k=1}^K \max_{m=1...M} \max(0, x_{kM+m} - y_{kM+m}) \tag{21}$$

## 4.2 Action Invariance ($\mathcal{I}$)

Invariance to the $\mathcal{I}$ backup operator in Eq. (7) gives the following update across $s, a \in \mathcal{S} \times \mathcal{A}$

$$d_\theta\big(\psi(s), \phi(s, a)\big) \leftarrow 0, \tag{22}$$

which can be directly enforced with the following loss across the batch:

$$\mathcal{L}_\mathcal{I}\big(\phi, \psi; \{s_i, a_i, s'_i, g_i\}_{i=1}^N\big) = \sum_{i,j=1}^N d_{\text{MRN}}\big(\psi(s_i), \phi(s_i, a_j)\big). \tag{23}$$

## 4.3 Temporal Invariance ($\mathcal{T}$)

Invariance to the $\mathcal{T}$ backup operator in Eq. (5) corresponds to the following update performed with respect to $\phi(s, a)$:

$$e^{-d_{\text{MRN}}(\phi(s,a), \psi(g))} \leftarrow \mathbb{E}_{\text{P}(s'|s,a)}\Big[e^{\log \gamma - d_{\text{MRN}}(\psi(s'), \psi(g))}\Big]. \tag{24}$$

This update is enforced by minimizing a divergence between the LHS and samples from the RHS expectation. Classic approaches for backups in deep RL include the $\ell_2$ distance to the target (RHS) [34], or when values can be interpreted as probabilities, a binary cross-entropy loss [35].

We use the following Bregman divergence [36], which we find empirically is more stable for learning the update in Eq. (24) (c.f. the Itakura-Saito distance [37] and Linex losses [38, 39]).

$$D_T(d, d') \triangleq \exp(d - d') - d. \tag{25}$$

We discuss this divergence and prove correctness in Appendix E. With the divergence, the $\mathcal{T}$-invariance loss is:

$$\mathcal{L}_\mathcal{T}\left(\phi, \psi; \{s_i, a_i, s_i', g_i\}_{i=1}^N\right) = \sum_{i=1}^N \sum_{j=1}^N D_T\left(d_{\mathrm{MRN}}(\phi(s_i, a_i), \psi(g_j)), d_{\mathrm{MRN}}(\psi(s_i'), \psi(g_j)) - \log\gamma\right) \tag{26}$$

We minimize this loss only with respect to $\phi$, stopping the gradient through $\psi$. This avoids the moving target that classically necessitates learning separate target networks in RL [34].

## 4.4 The Overall Distance Learning Objective

We can express the overall critic loss as:

$$\mathcal{L}_{\mathrm{TMD}}\left(\phi, \psi; \overline{\psi}, \mathcal{B}\right) = \mathcal{L}_{\mathrm{NCE}}\left(\phi, \psi; \mathcal{B}\right) + \zeta\left(\mathcal{L}_\mathcal{I}\left(\phi, \psi; \mathcal{B}\right) + \mathcal{L}_\mathcal{T}\left(\phi, \overline{\psi}; \mathcal{B}\right)\right) \tag{27}$$

$$\text{for batch} \quad \mathcal{B} \sim \pi_\beta = \{s_i, a_i, s_i', g_i\}_{i=1}^N$$

We minimize Eq. (27) with respect to $\phi$ and $\psi$, where $\overline{\psi}$ is a separate copy of the representation network $\psi$ (stop-gradient). Here, $\zeta$ controls the weight of the contrastive loss and invariance constraints, and batches are sampled

$$\{s_i, a_i, s_i', g_i\}_{i=1}^N \sim \pi_\beta,$$

for $s_i'$ the state following $s_i$, and $g_i$ the state $K$ steps ahead of $s_i$ for $K \sim \mathrm{Geom}(1 - \gamma)$. In theory, $\zeta^{-1}$ should be annealed between 1 at the start of training (to extract the distance $\mathcal{C}(\pi)$), toward 0 at the end of training to enforce invariance to $(\mathcal{T} \circ \mathcal{I})$, though in practice we find it suffices to keep $\zeta$ constant in the environments we tested.

In practice, we pick $\zeta$ based on how much stitching and stochasticity we expect in the environment — when $\zeta$ is large, we more aggressively try and improve on the initial distance $\mathcal{C}(\pi_\beta)$ describing the dataset policy $\pi_\beta$.

## 4.5 Policy Extraction

We finally extract the goal-conditioned policy $\pi(s, g) : \mathcal{S}^2 \to \mathcal{A}$ with the learned distance $d_\theta$:

$$\min_\pi \mathbb{E}_{\{s_i, a_i, s_i', g_i\}_{i=1}^N \sim \pi_\beta}\left[\sum_{i,j=1}^N d_\theta\left((s_i, \pi(s_i, g_j)), g_j\right)\right]. \tag{28}$$

For conservatism [40], we augment Eq. (28) with a behavioral cloning loss against $\pi_\beta$ via behavior-constrained deep deterministic policy gradient [41]. Using additional goals $g_i$ sampled from the same trajectory as $s_i$ in Eq. (28) could also be done through an extra tuned parameter (cf. Bortkiewicz et al. [42], Park et al. [43]). Denoting these hyperparameters as $\lambda$ and $\alpha$ respectively, the overall policy extraction objective is:

$$\min_\pi \mathbb{E}_{\{s_i, a_i, s_i', g_i\}_{i=1}^N \sim \pi_\beta}\left[\mathcal{L}_\pi\left(\pi; \phi, \psi, \{s_i, a_i, s_i', g_i\}_{i=1}^N\right)\right] \tag{29}$$

$$\mathcal{L}_\pi \triangleq \sum_{i,j=1}^N (1 - \lambda)d_{\mathrm{MRN}}\left(\phi(s_i, \hat{a}_{ij}), \psi(g_j)), \psi(g_j)\right) + \lambda d_{\mathrm{MRN}}\left(\phi(s_i, \hat{a}_{ii}), \psi(g_i)\right) + \alpha\left\|\hat{a}_{ii} - a_i\right\|_2^2$$

$$\text{where } \hat{a}_{ij} = \pi(s_i, g_j). \tag{30}$$

Prior offline RL methods use similar $\alpha$ and $\lambda$ hyperparameters, which must be tuned per environment [43].

Table 1: OGBench Evaluation

| Dataset | Methods | | | | | | |
|---|---|---|---|---|---|---|---|
| | **TMD** | CMD | CRL | QRL | GCBC | GCIQL | GCIVL |
| `humanoidmaze_medium_navigate` | **64.6**$^{(\pm 1.1)}$ | 61.1$^{(\pm 1.6)}$ | 59.9$^{(\pm 1.3)}$ | 21.4$^{(\pm 2.9)}$ | 7.6$^{(\pm 0.6)}$ | 27.3$^{(\pm 0.9)}$ | 24.0$^{(\pm 0.8)}$ |
| `humanoidmaze_medium_stitch` | 68.5$^{(\pm 1.7)}$ | **64.8**$^{(\pm 3.7)}$ | 36.2$^{(\pm 0.9)}$ | 18.0$^{(\pm 0.7)}$ | 29.0$^{(\pm 1.7)}$ | 12.1$^{(\pm 1.1)}$ | 12.3$^{(\pm 0.6)}$ |
| `humanoidmaze_large_stitch` | **23.0**$^{(\pm 1.5)}$ | 9.3$^{(\pm 0.7)}$ | 4.0$^{(\pm 0.2)}$ | 3.5$^{(\pm 0.5)}$ | 5.6$^{(\pm 1.0)}$ | 0.5$^{(\pm 0.1)}$ | 1.2$^{(\pm 0.2)}$ |
| `humanoidmaze_giant_navigate` | **9.2**$^{(\pm 1.1)}$ | 5.0$^{(\pm 0.8)}$ | 0.7$^{(\pm 0.1)}$ | 0.4$^{(\pm 0.1)}$ | 0.2$^{(\pm 0.0)}$ | 0.5$^{(\pm 0.1)}$ | 0.2$^{(\pm 0.1)}$ |
| `humanoidmaze_giant_stitch` | **6.3**$^{(\pm 0.6)}$ | 0.2$^{(\pm 0.1)}$ | 1.5$^{(\pm 0.5)}$ | 0.4$^{(\pm 0.1)}$ | 0.1$^{(\pm 0.0)}$ | 1.5$^{(\pm 0.1)}$ | 1.7$^{(\pm 0.1)}$ |
| `pointmaze_teleport_stitch` | 29.3$^{(\pm 2.2)}$ | 15.7$^{(\pm 2.9)}$ | 4.1$^{(\pm 1.1)}$ | 8.6$^{(\pm 1.9)}$ | 31.5$^{(\pm 3.2)}$ | 25.2$^{(\pm 1.0)}$ | **44.4**$^{(\pm 0.7)}$ |
| `antmaze_medium_navigate` | 93.6$^{(\pm 1.0)}$ | 92.4$^{(\pm 0.9)}$ | **94.9**$^{(\pm 0.5)}$ | 87.9$^{(\pm 1.2)}$ | 29.0$^{(\pm 1.7)}$ | 12.1$^{(\pm 1.1)}$ | 12.3$^{(\pm 0.6)}$ |
| `antmaze_large_navigate` | 81.5$^{(\pm 1.7)}$ | **84.1**$^{(\pm 2.1)}$ | 82.7$^{(\pm 1.4)}$ | 74.6$^{(\pm 2.3)}$ | 24.0$^{(\pm 0.6)}$ | 34.2$^{(\pm 1.3)}$ | 15.7$^{(\pm 1.9)}$ |
| `antmaze_large_stitch` | **37.3**$^{(\pm 2.7)}$ | 29.0$^{(\pm 2.3)}$ | 10.8$^{(\pm 0.6)}$ | 18.4$^{(\pm 0.7)}$ | 3.4$^{(\pm 1.0)}$ | 7.5$^{(\pm 0.7)}$ | 18.5$^{(\pm 0.8)}$ |
| `antmaze_teleport_explore` | **49.6**$^{(\pm 1.5)}$ | 0.2$^{(\pm 0.1)}$ | 19.5$^{(\pm 0.8)}$ | 2.3$^{(\pm 0.7)}$ | 2.4$^{(\pm 0.4)}$ | 7.3$^{(\pm 1.2)}$ | 32.0$^{(\pm 0.6)}$ |
| `antmaze_giant_stitch` | **2.7**$^{(\pm 0.6)}$ | 2.0$^{(\pm 0.5)}$ | 0.0$^{(\pm 0.0)}$ | 0.4$^{(\pm 0.2)}$ | 0.0$^{(\pm 0.0)}$ | 0.0$^{(\pm 0.0)}$ | 0.0$^{(\pm 0.0)}$ |
| `scene_noisy` | 19.6$^{(\pm 1.7)}$ | 4.0$^{(\pm 0.7)}$ | 1.2$^{(\pm 0.3)}$ | 9.1$^{(\pm 0.7)}$ | 1.2$^{(\pm 0.2)}$ | 25.9$^{(\pm 0.8)}$ | **26.4**$^{(\pm 1.7)}$ |
| `visual_antmaze_teleport_stitch` | **38.5**$^{(\pm 1.5)}$ | 36.0$^{(\pm 2.1)}$ | 31.7$^{(\pm 3.2)}$ | 1.4$^{(\pm 0.8)}$ | 31.8$^{(\pm 1.5)}$ | 1.0$^{(\pm 0.2)}$ | 1.4$^{(\pm 0.4)}$ |
| `visual_antmaze_large_stitch` | **26.6**$^{(\pm 2.8)}$ | 8.1$^{(\pm 1.3)}$ | 11.1$^{(\pm 1.3)}$ | 0.6$^{(\pm 0.3)}$ | 23.6$^{(\pm 1.4)}$ | 0.1$^{(\pm 0.0)}$ | 0.8$^{(\pm 0.3)}$ |
| `visual_antmaze_giant_navigate` | 40.1$^{(\pm 2.6)}$ | 37.3$^{(\pm 2.4)}$ | **47.2**$^{(\pm 0.9)}$ | 0.1$^{(\pm 0.1)}$ | 0.4$^{(\pm 0.1)}$ | 0.1$^{(\pm 0.2)}$ | 1.0$^{(\pm 0.4)}$ |
| `visual_cube_triple_noisy` | 17.7$^{(\pm 0.7)}$ | 16.1$^{(\pm 0.7)}$ | 15.6$^{(\pm 0.6)}$ | 8.6$^{(\pm 2.1)}$ | 16.2$^{(\pm 0.7)}$ | 12.5$^{(\pm 0.6)}$ | **17.9**$^{(\pm 0.5)}$ |

We **bold** the best performance. Success rate (%) is presented with the standard error across six seeds. All datasets contain 5 separate tasks each. We record the aggregate across all 5 tasks.

## 5 Experiments

In our experiments, we evaluate the performance of TMD on tasks from the OGBench benchmark [43]. We aim to answer the following questions:

1. Do the invariance terms in Eq. (6) improve performance quantitatively in offline RL settings?

2. Is the contrastive loss in Eq. (19) necessary to facilitate learning these tasks?

3. What capabilities does TMD enable for compositional task learning?

### 5.1 Experimental Results

We evaluate TMD across evaluation tasks in OG-Bench for the environments and datasets listed in Table 1. The experiments use 6 seeds in all environments, and report the success rates aggregated across the 5 evaluation tasks (goals) provided with each environment. Of particular interest are the "`teleport`" and "`stitch`" environments, which respectively test the ability to handle stochasticity and composition.

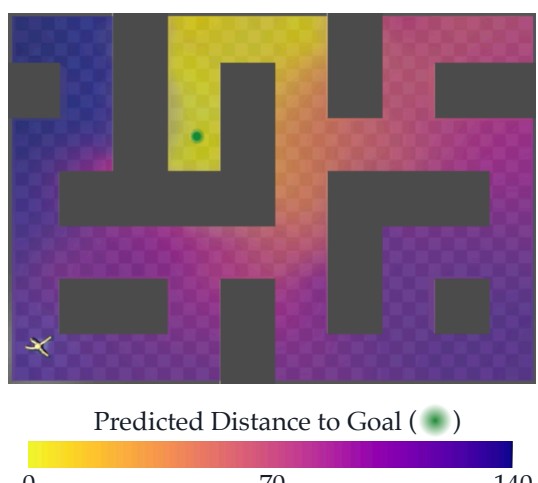

Predicted Distance to Goal (   )

0          70          140

Figure 3: An example distance heatmap learned by TMD in `pointmaze_large_stitch`. Darker colors indicate larger distances.

We compare against the Goal-Conditioned Behavioral Cloning (GCBC), Goal-Conditioned Implicit Q-Learning (GCIQL), Goal-Conditioned Implicit Value Learning (GCIVL), Contrastive Reinforcement Learning (CRL), and Quasimetric Reinforcement Learning (QRL) algorithms, using the reference results provided by OGBench [43]. We implement and evaluate Contrastive Metric Distillation (CMD) [16], which also learns a quasimetric temporal distance, but does not enforce the constraint of $\mathcal{T}$ or $\mathcal{I}$ invariance and uses a separate critic architecture. GCBC uses imitation learning to learn a policy that follows the given trajectories within a dataset [44]. CRL [15] performs policy improvement by fitting a value function via contrastive learning. QRL [18] learns a quasimetric value function to recover optimal

distances in deterministic settings. GCIQL and GCIVL use expectile regression to fit a value function [45].

TMD consistently outperforms QRL and CRL in the stitching environments. In the stochastic `teleport` environments, TMD outperforms both CRL and QRL by a considerable margin — in `pointmaze_teleport_stitch` TMD outperforms CRL and QRL by over 3x. An example distance learned by TMD in `antmaze_large_stitch` is visualized as a heatmap in Fig. 3.

## 5.2 Ablation Study

We perform an ablation study on the `pointmaze_teleport_stitch` environment to evaluate the importance of the invariance terms and the contrastive initialization loss in TMD. We separately disable the contrastive, $\mathcal{T}$ invariance, and $\mathcal{I}$ invariance component during training and observe its effects. We also examine the empirical effects of stopping gradients when calculating $\mathcal{L}_{\mathcal{T}}$. We log the corresponding success rate for each of the ablations in 4.

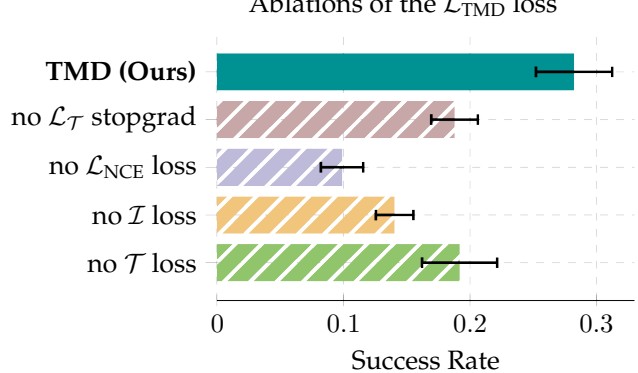

Figure 4: We ablate the loss components of TMD in the `pointmaze_teleport_stitch` environment.

Our ablation studies answer questions 2 and 3, in which we demonstrate that by removing some of the invariances or removing the contrastive loss, the performance of TMD decreases to levels similar to CRL and QRL. Similarly, we see the importance of keeping the contrastive objective, as the performance of TMD degrades even more despite the presence of other loss components. We also note the empirical performance of TMD is better when we stop gradients on $\mathcal{L}_{\mathcal{T}}$. We provide further ablation details in Appendix D.2.

## 6 Discussion

In this work, we introduce Temporal Metric Distillation (TMD), an offline goal-conditioned reinforcement learning method that learns representations which exploit the quasimetric structure of temporal distances. Our approach unifies quasimetric, temporal-difference, and Monte Carlo learning approaches to GCRL by enforcing a set of invariance properties on the learned distance function. To the best of our knowledge, TMD is the first method that can exploit the quasimetric structure of temporal distances to learn optimal policies from offline data, even in stochastic settings (see Fig. 2). On a standard suite of offline GCRL benchmarks, TMD outperforms prior methods, in particular on long-horizon tasks that require stitching together trajectories across noisy dynamics and visual observations.

### 6.1 Limitations and Future Work

Future work could examine more principled ways to set the $\zeta$ parameter in our method, or if there are ways to more directly integrate the contrastive and invariance components of the loss function. Future work could also explore integrating the policy extraction objective more directly into the distance learning to enable desirable properties (stitching through architecture, horizon generalization) at the level of the policy. While we used the MRN [17] architecture in our experiments, alternative architectures such as Interval Quasimetric Embedding (IQE) [20] that enforce the triangle inequality could be more expressive. While the size of models studied in our experiments make them unlikely to pose any real-world risks, methods which implicitly enable long-horizon decision making could have unintended consequences or poor interpretability. Future work should consider these implications.

**Acknowledgements**

We would like to thank Seohong Park, Qiyang (Colin) Li, Catherine Ji, Cameron Allen, and Kyle Stachowicz for relevant discussions and feedback on this work. This research was partly supported by AFOSR FA9550-22-1-0273, DARPA TIAMAT, and the DoD NDSEG fellowship.

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

## A  Code

Code and videos can be found at `https://tmd-website.github.io/`. The evaluation and base agent structure follows the OGBench codebase [43]. The TMD agent is implemented in `https://github.com/vivekmyers/tmd-release/blob/master/impls/agents/tmd.py`.

## B  Analysis of TMD

This section provides the proofs of the results in Section 3.5. The main result is Theorem 3, which shows that enforcing the TMD constraints on a learned quasimetric distance recovers the optimal distance $d_{\text{SD}}^*$.

**Theorem 1.** *Take $d \in \mathcal{D}$ and consider the sequence*

$$d_n = \mathcal{P}^n(d).$$

*Then, $d_n$ converges uniformly to a fixed point $d_\infty \in \mathcal{Q}$.*

*Proof.* From Lemma 7, we have that $d_{n+1}(s,g) \leq d_n(s,g)$ for all $s, g \in \mathcal{S}$. Thus, the sequence $\{d_n\}$ is monotonically decreasing (and positive). By the monotone convergence theorem, the sequence converges pointwise to a limit $d_\infty$. Since $\mathcal{S}$ is compact, by Dini's theorem [46], the convergence is uniform, i.e., $d_n \to d_\infty$ under the $L^\infty$ topology over $\mathcal{D}$.

To see that $d_\infty$ is a fixed point of $\mathcal{P}$, we note that if $\mathcal{P}d_\infty = d' \neq d_\infty$, we can construct disjoint neighborhoods $N$ of $d_\infty$ and $N'$ of $d'$ (since $L^\infty(\mathcal{D})$ is normed vector space and thus Hausdorff). By construction, the preimage $\mathcal{P}^{-1}(N')$ contains $d_\infty$ and is open by Lemma 6. Thus, we can define another, smaller open neighborhood $N'' = N \cap \mathcal{P}^{-1}(N')$ of $d_\infty$. Now, since $d_n \to d_\infty$, there exists some $k$ so $d_k, d_{k+1} \in N'' \subset N$. But then since $d_k \in \mathcal{P}^{-1}(N')$, we have that $d_{k+1} \in N'$. This is a contradiction as $N$ and $N'$ were disjoint by construction.

Thus, we have that $d_\infty$ is a fixed point of $\mathcal{P}$. That $d_\infty \in \mathcal{Q}$ follows from Lemma 5. $\qquad\square$

**Theorem 3.** *The TMD algorithm converges pointwise to the optimal successor distance $d_{\text{SD}}^*$ for any policy $\pi$ with full state and action coverage, i.e.,*

$$\lim_{n \to \infty} (\mathcal{P}_* \circ \mathcal{T} \circ \mathcal{I})^n \, \mathcal{C}(\pi) = d_{\text{SD}}^*. \tag{17}$$

*Proof of Theorem 3.* The initial distance $\mathcal{C}(\pi) = d_{\text{SD}}{}^\pi \geq d_{\text{SD}}^*$ for any policy $\pi$. So, $\mathcal{C}(\pi) \in \mathcal{D}_+^*$. Define the sequence of distances $d_n = (\mathcal{P}_* \circ \mathcal{T} \circ \mathcal{I})^n \mathcal{C}(\pi)$ in $\mathcal{D}_+^*$. Note that $\mathcal{P}_*$ and $\mathcal{I}$ are monotone decreasing. So, the restriction $(d_n)_{|\mathcal{X}}$ is monotonically decreasing on the domain $\mathcal{X} = \mathcal{S} \times (\mathcal{S} \cup \mathcal{S} \times \mathcal{A})$, and thus converges pointwise on $\mathcal{X}$ as $n \to \infty$.

Since $\mathcal{T}$ and $\mathcal{P}_*$ are continuous operators (Lemma 6 and Eq. (5)), and $\mathcal{T}$ is fully-determined by the restriction to $\mathcal{X}$, the sequence $(\mathcal{P} \circ \mathcal{T})d_n = d_{n+1}$ converges pointwise on its full domain. The pointwise limit of $d_n$ is a fixed point of $(\mathcal{P}_* \circ \mathcal{T} \circ \mathcal{I})$, which must be the unique fixed point $d_{\text{SD}}^*$ on $\mathcal{D}_+^*$ by Lemma 4. □

**Lemma 4.** *The optimal successor distance $d_{\text{SD}}^*$ is the unique fixed point of $\mathcal{P}_* \circ \mathcal{T} \circ \mathcal{I}$ on $\mathcal{D}_+^*$.*

*Proof of Lemma 4.* For existence, we note

$$
\begin{aligned}
(\mathcal{P}_* \circ \mathcal{T} \circ \mathcal{I})d_{\text{SD}}^* &= (\mathcal{P}_* \circ \mathcal{T})(\mathcal{I}d_{\text{SD}}^*) && \text{(Remark 2)} \\
&= (\mathcal{P}_* \circ \mathcal{T})d_{\text{SD}}^* && \text{(Bellman optimality of } Q_g^*) \\
&= \mathcal{P}_* d_{\text{SD}}^*. && \text{(Lemma 5)} \\
&= d_{\text{SD}}^*. && (31)
\end{aligned}
$$

For uniqueness, we need to show that $(\mathcal{P}_* \circ \mathcal{T} \circ \mathcal{I})$ has no fixed points besides $d_{\text{SD}}^*$ in $\mathcal{D}_+^*$. Suppose there exists some $d \in \mathcal{D}_+^*$ such that $(\mathcal{P}_* \circ \mathcal{T} \circ \mathcal{I})d = d$. Then, we have for $x \in \mathcal{S} \cup \mathcal{S} \times \mathcal{A}$, $s, g \in \mathcal{S}$, and $a \in \mathcal{A}$:

$$(\mathcal{P}_* \circ \mathcal{T} \circ \mathcal{I})d\big(x, (g, a)\big) = d\big(x, (g, a)\big) = d(x, g). \tag{32}$$

Denote by $Q(s, a) = e^{-d((s,a),g)}$, and let $\mathcal{B}$ be the goal-conditioned Bellman operator defined as

$$\mathcal{B}Q(s, a) \triangleq \mathbb{E}_{P(s'|s,a)}\big[\mathbb{1}\{s' = g\} + \gamma Q(s', g)\big] \tag{33}$$

At any fixed point $d \in \mathcal{D}_+^*$, we have

$$(\mathcal{P}_* \circ \mathcal{T} \circ \mathcal{I})d\big((s, a), g\big) = d\big((s, a), g\big) \tag{34}$$

This last expression implies that

$$
\begin{aligned}
Q(s, a) &= \exp\big[-d\big((s, a), g\big)\big] \\
&= \exp\big[-(\mathcal{P}_* \circ \mathcal{T} \circ \mathcal{I})d\big((s, a), g\big)\big] \\
&\leq \mathbb{E}_{P(s'|s,a)}\big[\min_{a' \in \mathcal{A}} \exp d\big((s', a'), g\big)\big] - \log \gamma \\
&= \mathcal{B}Q(s, a). && (35)
\end{aligned}
$$

Since $\mathcal{B}$ is a contraction on the exponentiated distance space, and $d\big((s, a), g\big) \geq d_{\text{SD}}^*\big((s, a), g\big)$, Eq. (35) is only consistent with $Q(s, a) = Q_g^*(s, a)$. This implies that

$$d\big((s, a), g\big) = d_{\text{SD}}^*\big((s, a), g\big). \tag{36}$$

We also know that at this fixed point, $d(s, (s, a)) = 0$, and thus from Eq. (36) we have

$$d(s, g) = d_{\text{SD}}^*(s, g). \tag{37}$$

So, $d = d_{\text{SD}}^*$ must be the unique fixed point of $(\mathcal{P}_* \circ \mathcal{T} \circ \mathcal{I})$.

□

# C  Path Relaxation and Quasimetric Distances

We provide short proofs of the claims in 

**Lemma 5.** *We have $\mathcal{P}(d) = d$ if and only if $d \in \mathcal{Q}$.*

*Proof.*
$$\begin{aligned}
\mathcal{P}(d)(s, g) &= \min_{w \in \mathcal{S}} \big[ d(s, w) + d(w, g) \big] \\
&\leq d(s, s) + d(s, g) \\
&= d(s, g).
\end{aligned}$$
$\square$

**Lemma 6.** *The path relaxation operator $\mathcal{P}$ is continuous with respect to the $L^\infty$ topology over $\mathcal{D}$.*

*Proof.* Let $d, d' \in \mathcal{D}$ and $\epsilon > 0$. We have

$$\begin{aligned}
\big| \mathcal{P}(d)(s, g) - \mathcal{P}(d')(s, g) \big| &= \bigg| \min_{w \in \mathcal{S}} \big[ d(s, w) + d(w, g) \big] - \min_{w \in \mathcal{S}} \big[ d'(s, w) + d'(w, g) \big] \bigg| \\
&\leq \min_{w \in \mathcal{S}} \big| d(s, w) + d(w, g) - d'(s, w) - d'(w, g) \big| \\
&\leq \min_{w \in \mathcal{S}} \big| d(s, w) - d'(s, w) \big| + \min_{w \in \mathcal{S}} \big| d(w, g) - d'(w, g) \big| \\
&\leq \| d - d' \|_\infty + \| d - d' \|_\infty \\
&= 2 \| d - d' \|_\infty.
\end{aligned}$$

Thus, if $\| d - d' \|_\infty < \epsilon/2$, we have $\big\| \mathcal{P}(d) - \mathcal{P}(d') \big\|_\infty < \epsilon$.
$\square$

**Lemma 7.** *For any $s, g \in \mathcal{S}$ and $d \in \mathcal{D}$ we have that $\mathcal{P}(d)(s, g) \leq d(s, g)$.*

*Proof.* Let $d, d' \in \mathcal{D}$ and $\epsilon > 0$. We have

$$\begin{aligned}
\big| \mathcal{P}(d)(s, g) - \mathcal{P}(d')(s, g) \big| &= \bigg| \min_{w \in \mathcal{S}} \big[ d(s, w) + d(w, g) \big] - \min_{w \in \mathcal{S}} \big[ d'(s, w) + d'(w, g) \big] \bigg| \\
&\leq \min_{w \in \mathcal{S}} \big| d(s, w) + d(w, g) - d'(s, w) - d'(w, g) \big| \\
&\leq \min_{w \in \mathcal{S}} \big| d(s, w) - d'(s, w) \big| + \min_{w \in \mathcal{S}} \big| d(w, g) - d'(w, g) \big| \\
&\leq \| d - d' \|_\infty + \| d - d' \|_\infty \\
&= 2 \| d - d' \|_\infty.
\end{aligned}$$

Thus, if $\| d - d' \|_\infty < \epsilon/2$, we have $\big\| \mathcal{P}(d) - \mathcal{P}(d') \big\|_\infty < \epsilon$.
$\square$

**Lemma 5.** *We have $\mathcal{P}(d) = d$ if and only if $d \in \mathcal{Q}$.*

*Proof.* ($\Rightarrow$) Suppose $\mathcal{P}(d) = d$. Then, for all $s, g, w \in \mathcal{S}$ we have
$$d(s, g) = \mathcal{P}(d)(s, g) = \min_{w \in \mathcal{S}} \big[ d(s, w) + d(w, g) \big] \leq d(s, w) + d(w, g).$$

Thus, $d \in \mathcal{Q}$.

($\Leftarrow$) Suppose $d \in \mathcal{Q}$. Then, for all $s, g \in \mathcal{S}$ we have
$$d(s, g) \leq \min_{w \in \mathcal{S}} \big[ d(s, w) + d(w, g) \big] = \mathcal{P}(d)(s, g).$$

We also have $\mathcal{P}(d)(s, g) \leq d(s, g)$ by . Thus, $\mathcal{P}(d) = d$.
$\square$

Table 2: Hyperparameters for TMD

| Hyperparameter | Value |
|---|---|
| batch size | 256 |
| learning rate | $3 \cdot 10^{-4}$ |
| discount factor | 0.995 |
| invariance weight $\zeta$ | 0.01 in `medium` locomotion environments, 0.1 otherwise |

Table 3: Network configuration for TMD.

| Configuration | Value |
|---|---|
| latent dimension size | 512 |
| encoder MLP dimensions | $(512, 512, 512)$ |
| policy MLP dimensions | $(512, 512, 512)$ |
| layer norm in encoder MLPs | `True` |
| visual encoder (`visual-` envs) | `impala-small` |
| MRN components | 8 |
| $\mathcal{T}$ weighting on diagonal elements | 1 (navigation, play) |
| | 0.5 (stitch, explore, noisy) |

# D    Experimental Details

General hyperparameters are provided in Table 2.

We implemented TMD using JAX [47] within the OGBench [43] framework. OGBench requires a per-environment hyperparameter $\alpha$ controlling the behavioral cloning weight to be tuned for each method based on the scale of its losses. We generally found TMD to work well with similar $\alpha$ values to those used by CRL. We used the same values of $\alpha$ as CMD's implementation. For a complete list of alpha values, please refer to the code release of the paper.

To prevent gradients from overflowing, we clip the $\mathcal{T}$ invariance loss per component to be no more than 5. We also found using a slightly smaller batch size of 256 compared to 512 to be helpful for reducing memory usage.

## D.1    Implementation Details

The network architecture for TMD is described in Table 3. The "MRN components" refers to the number of ensemble terms $K$ in Eq. (21). We found $K = 8$ components enabled stable learning and expressive distances. We weigh the off-diagonal element, corresponding to the product of the marginals $p(s)p(g)$, on a 0-1 scale compared to the diagonal elements, corresponding to the joint distribution $p(s, g)$. A scale of 0 corresponds to the off-diagonal elements weighing the same as the diagonal elements, and a scale of 1 means that only diagonal elements will matter for $\mathcal{T}$-operator.

## D.2    Ablations

The full ablation results for TMD in the `pointmaze-teleport-stitch` are presented in Table 4 with success rates and standard errors.

Table 4: Ablation Success rate.

| Ablation | Success Rate |
|---|---|
| None | $\mathbf{29.3}^{(\pm \mathbf{2.2})}$ |
| No gradient stopping in $\mathcal{L}_{\mathcal{T}}$ | $18.7^{(\pm 1.8)}$ |
| No contrastive loss | $9.8^{(\pm 1.7)}$ |
| No $\mathcal{I}$ loss | $13.3^{(\pm 2.9)}$ |
| No $\mathcal{T}$ loss | $18.5^{(\pm 2.1)}$ |

### D.3 Computational Resources

Experiments were run using NVIDIA A6000 GPUs with 48GB of memory, and 4 CPU cores and 1 GPU per experiment. Each state-based experiment took around 2 hours to run with these resources, and each pixel-based experiment took around 4 hours.

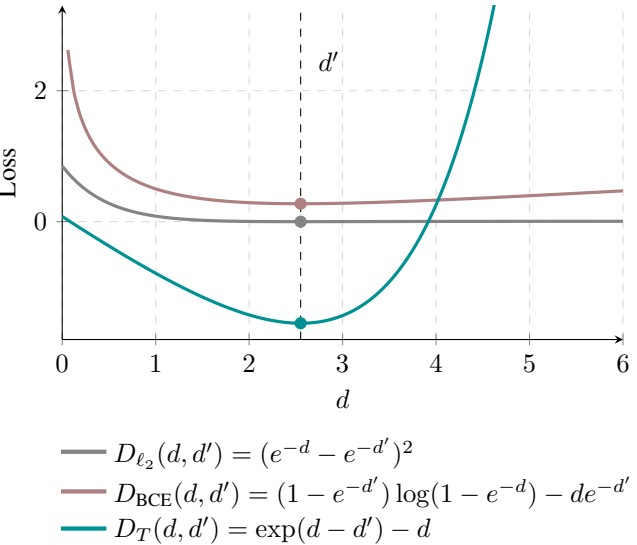

Figure 5: Comparison of Bregman divergences for $e^{-d}$ onto $e^{-d'}$ in expectation. All losses are minimized at $d = d'$, and share the property that they will be minimized in expectation when $e^{-d} = \mathbb{E}[e^{-d'}]$. But only the $D_T(d, d')$ loss has non-vanishing gradients $d \gg d'$ for large $d'$.

## E   Bregman Divergence in $\mathcal{T}$-invariance

Recall the divergence used in Eq. (25):

$$D_T(d, d') \triangleq \exp(d - d') - d. \tag{25}$$

This divergence is proportional to the Bregman divergence [36] for the function $F(x) = -\log(x)$, similar to the Itakura-Saito divergence [37].

$$
\begin{aligned}
D_F(e^{-d'}, e^{-d}) &= F(e^{-d'}) - F(e^{-d}) - F'(e^{-d})(e^{-d'} - e^{-d}) \\
&= d' - d + \frac{1}{e^{-d}}(e^{-d'} - e^{-d}) \\
&= d' - d + \exp(d - d') - 1 \\
&= D_T(d, d') + d' - 1.
\end{aligned}
\tag{38}
$$

The minimizer of Eq. (25) satisfies

$$\underset{d \geq 0}{\arg\min} \, \mathbb{E}_{d'}[D_T(d, d')] = -\log \mathbb{E}_{d'}[e^{-d'}] \tag{39}$$

when $d'$ is a random "target" distance [48]. In other words, using Eq. (25) as a loss function regresses $e^{-d}$ onto the expected value of $e^{-d'}$ (or onto the expected value of $e^{\log \gamma - d'}$ as used in Eq. (26)).

The key advantage of this divergence when backing up temporal distances is that the gradients do not vanish when either $d$, $d'$, or the difference between them is small or large. This property is *not* shared by more standard loss functions like the squared loss or binary cross-entropy loss when applied to the probability space and the models (distances) are in log-probability space.

---

**Algorithm 1:** Temporal Metric Distillation (TMD)

---

1: **input:** dataset $\mathcal{D}$, learning rate $\eta$
2: initialize representations $\phi, \psi$, policy $\pi$
3: **while** training **do**
4:      sample $\mathcal{B} = \{s_i, a_i, s'_i, g_i\}_{i=1}^N \sim \mathcal{D}$
5:      $\overline{\psi} \leftarrow \psi$
6:      $(\phi, \psi) \leftarrow (\phi, \psi) - \eta \nabla_{\phi,\psi} \mathcal{L}_{\text{TMD}}(\phi, \psi; \overline{\psi}, \mathcal{B})$             ▷ *Eq. (27)*
7:      $\pi \leftarrow \pi - \eta \nabla_\pi \mathcal{L}_\pi(\phi, \psi, \pi; \mathcal{B})$                       ▷ *Eq. (28)*
8: **return** $\pi$

---

Table 5: Ablation of $\mathcal{T}$-invariance loss in `antmaze-teleport-stitch`

| Loss | Success Rate |
|---|---|
| $D_T$ **(Ours)** | $\mathbf{29.3}^{(\pm 2.2)}$ |
| $D_{\ell_2}$ | $16.1^{(\pm 1.9)}$ |
| $D_{\text{BCE}}$ | $15.1^{(\pm 1.9)}$ |

### E.1 Empirical Comparison

In practice, we found it was important to use this divergence in TMD for stable learning (Table 5). Ths loss could also be applied to other GCRL algorithms where learned value functions are probabilities but are predicted in log-space to improve gradients. Future work should explore this divergence in other GCRL algorithms to improve training compared to the more commonly used squared loss or binary cross-entropy loss [35].

## F Algorithm Pseudocode

Full pseudocode for TMD is provided in Algorithm 1. We provide the full TMD loss function in Eq. (27) and the policy extraction loss in Eq. (28) below for reference:

$$\mathcal{L}_{\text{TMD}}(\phi, \psi; \overline{\psi}, \mathcal{B}) = \mathcal{L}_{\text{NCE}}(\phi, \psi; \mathcal{B}) + \zeta\Big(\mathcal{L}_{\mathcal{I}}(\phi, \psi; \mathcal{B}) + \mathcal{L}_{\mathcal{T}}(\phi, \overline{\psi}; \mathcal{B})\Big) \tag{27}$$

$$\mathcal{L}_\pi\big(\pi; \phi, \psi, \{s_i, a_i, s'_i, g_i\}_{i=1}^N\big) = \sum_{i,j=1}^N (1-\lambda) d_{\text{MRN}}\big(\phi(s_i, \hat{a}_{ij}), \psi(g_j)\big), g_j\big)$$

$$+ \lambda d_{\text{MRN}}\big(\phi(s_i, \hat{a}_{ii}), \psi(g_i)\big) + \alpha \big\|\hat{a}_{ii} - a_i\big\|_2^2 \tag{28}$$

where $\hat{a}_{ij} = \pi(s_i, g_j)$, batch $\mathcal{B} \sim p^{\pi_\beta} = \{s_i, a_i, s'_i, g_i\}_{i=1}^N$.

The components of Eq. (27) are (see Section 4):

$$\mathcal{L}_{\text{NCE}}\big(\phi, \psi; \{s_i, a_i, s'_i, g_i\}_{i=1}^N\big) = \sum_{i=1}^N \log\left(\frac{e^{f(s_i, a_i, g_i)}}{\sum_{j=1}^N e^{f(s_j, a_j, g_i)}}\right) \tag{19}$$

$$\mathcal{L}_{\mathcal{I}}\big(\phi, \psi; \{s_i, a_i, s'_i, g_i\}_{i=1}^N\big) = \sum_{i,j=1}^N d_{\text{MRN}}\big(\psi(s_i), \phi(s_i, a_j)\big) \tag{23}$$

$$\mathcal{L}_{\mathcal{T}}\big(\phi, \psi; \{s_i, a_i, s'_i, g_i\}_{i=1}^N\big) = \sum_{i,j=1}^N D_T\big(d_{\text{MRN}}(\phi(s_i, a_i), \psi(g_j)), d_{\text{MRN}}(\psi(s'_i), \psi(g_j)) - \log\gamma\big). \tag{26}$$

## Glossary

$Q^\beta$   The behavioral $Q$-function under policy $\pi_\beta$. 4

## Acronyms

**CMD** Contrastive Metric Distillation. 9, 16

**CRL** Contrastive Reinforcement Learning. 9, 10, 16, 19

**GCBC** Goal-Conditioned Behavioral Cloning. 9

**GCIQL** Goal-Conditioned Implicit Q-Learning. 9, 10

**GCIVL** Goal-Conditioned Implicit Value Learning. 9, 10

**GCRL** Goal-Conditioned Reinforcement Learning. 2, 3, 10, 18

**IQE** Interval Quasimetric Embedding. 10

**MLP** Multi-Layer perceptron. 16

**MRN** Metric Residual Network. 6, 10, 16

**QRL** Quasimetric Reinforcement Learning. 9, 10

**RL** Reinforcement Learning. 1, 2, 4, 8, 9

**TD** Temporal Difference. 1, 3, 4

**TMD** Temporal Metric Distillation. 2, 4–6, 9, 10, 13, 14, 16, 18

