# OpenReview forum: "Offline Goal-conditioned Reinforcement Learning with Quasimetric Representations"
_NeurIPS.cc/2025/Conference — NeurIPS 2025 poster_

### Official Review · Reviewer_CFjL · 2025-06-07

**Clarity:** 3
**Significance:** 3
**Originality:** 2
**Rating:** 5
**Confidence:** 4

**Summary:**

The paper proposes TMD, an offline GCRL method using temporal distance representations.
TMD aims to recover an optimal temporal distance by enforcing three constraints: (1) triangle inequality (enforced by quasimetric), (2) action invariance, (3) SARSA-like distance constraint ($d((s,a), s’) = -\log \gamma$).
The paper includes a theoretical analysis showing that, under these constraints, TMD recovers an optimal successor distance.
Empirically, TMD demonstrates strong performance on OGBench, outperforming other offline GCRL methods in 6 out of 8 tasks.

**Questions:**

**Q1. Final formula for modified successor distance**

* Could you clarify the motivation behind the final formula involving the pair y=(g,a)?
* I couldn't locate a corresponding discussion or derivation in [1].

**Q2. Visualization of learned Q-values**

* It would be helpful to include a visualization of the learned Q-values or successor distances (similar to Figure 2 in [1]) to better understand the learned representations.

**Q3. Evaluation in stochastic environments**
* Given TMD’s design to handle stochastic transitions, could you provide results for environments with added transition noise or inherent stochasticity?

**Clarifications & Minor typos**
* (Eq 7) Does $K \sim \gamma$ mean $K \sim Geom(1-\gamma)$?
* (Eq 24) If we are using Itakura-Saito distance, shouldn’t it be exp(d-d’) - (d-d’) - 1? Or is it because we are not using the gradient of d’?
* (Eq 28) \phi(s_i, \pi(s_i)) -> \phi(s_i, \pi(s_i, g_j))
* (L128) P
* (L135) 11.5
* (L266) in 3 -> in Figure 3

**Ethical Concerns:**

["NO or VERY MINOR ethics concerns only"]

**Final Justification:**

All of the questions were clarified, especially the question with the stochastic dynamics.
While writing of the paper is concerning, I think it can be improved without requiring major changes (e.g., without introducing new concepts or sections) in the final manuscript.

Thus, I recommend accepting this paper.

**Limitations:**

yes

**Quality:**

3

**Strengths And Weaknesses:**

**Strengths**

**S1. Strong Experimental results**
* TMD achieves state-of-the-art performance on OGBench, a challenging benchmark for offline GCRL, outperforming all other methods on 6 out of 8 tasks.

**S2. Solid theoretical support**
* The paper provides a theoretical justification that the three proposed constraints lead to an optimal successor distance metric (assuming full state-action coverage by the policy)
* Also, the SARSA-like formulation of the method makes TMD easier to use.

**S3. Ablation studies**
* The ablation study clearly shows the contribution of each component of TMD.
* The ablation study also provides insights on details of TMD (e.g., utilizing Bregman divergence in the log-space, stop-gradient)

&nbsp;

**Weaknesses:**

**W1. Slight mismatch in evaluation**
* TMD is designed to support stochastic transitions, which is an important advantage over QRL.
* However, the OGBench environments used in evaluation is deterministic (please correct me if it’s wrong).
* This raises questions about whether the stochasticity-aware aspects of TMD are being fully utilized.

---

> ### Author Rebuttal · Authors · 2025-07-31
>
> Thank you for your thoughtful comments and feedback. We will address some of your concerns below. Please let us know if you have any additional questions.
>
> > TMD is designed to support stochastic transitions, which is an important advantage over QRL. However, the OGBench environments used in evaluation is deterministic (please correct me if it's wrong).
>
> > Q3. Evaluation in stochastic environments
>
> The `teleport` environments feature portals that transport the agent to random locations. This makes them ideal for studying nontrivial stochastic dynamics.
>
> > Q1. Final formula for modified successor distance. Could you clarify the motivation behind the final formula involving the pair y=(g,a)? I couldn't locate a corresponding discussion or derivation in [1].
>
> Unlike the successor distance in CMD [1], we learn a distance over both states $s$ and state-action pairs $(s,a)$. This lets us reason about both ''Q'' values $d((s,a),g)$ and ''V'' values $d(s,g)$. Because the distance is a function over $((S \\times A) \\cup S) \\times ((S \\times A) \\cup S)$, we also need a sensible definition of the distance to a goal-action pair, $d(x, (g,a))$. This is what Eq. (7) provides—for a policy $\\pi$, the distance under $d^\\pi\_{SD}$ from $x \\to (g,a)$ is the distance from $x \\to g$ plus $-\\log (a \\mid g)$.
>
> We have revised the paper to clarify that we are extending the definition in [1] to handle this case.
>
> > Q2. Visualization of learned Q-values. It would be helpful to include a visualization of the learned Q-values or successor distances (similar to Figure 2 in [1]) to better understand the learned representations.
>
> We have generated a heatmap of successor distances for the antmaze-large-stitch environment. We will include this as an additional figure in the revised paper.
>
> #### Clarifications & Minor typos
>
> > (Eq 7) Does $K \\sim \\gamma$  mean $K \\sim \\operatorname{Geom}(1-\\gamma)$?
>
> Yes, we will fix this.
>
> > (Eq 24) If we are using Itakura-Saito distance, shouldn't it be exp(d-d')—(d-d')—1? Or is it because we are not using the gradient of d'?
>
> Correct, we can drop that term since we are minimizing the divergence wrt d.
>
> > (Eq 28), (L128), (L135), (L266)
>
> Thank you for noticing these. We will correct these errors in the revision.
>
> ---
>
> [1] Myers, V. et al., 2024. ''Learning Temporal Distances: Contrastive Successor Features Can Provide a Metric Structure for Decision-Making.'' ICML

---

> > ### Comment · Reviewer_CFjL · 2025-08-05
> >
> > Thank you for the clarifications.
> >
> > All of the questions were clarified, especially the question with the stochastic dynamics.

---

### Official Review · Reviewer_BR73 · 2025-06-16

**Clarity:** 2
**Significance:** 2
**Originality:** 2
**Rating:** 4
**Confidence:** 3

**Summary:**

I understand this work to be a follow-up of [1], which is in turn a follow-up of [2], which again can be seen as a follow-up of previous works such as [3] and [4], etc.

The offline setting prohibits direct access to the exploration policy.

======


[1] Learning Temporal Distances: Contrastive Successor Features Can Provide a Metric Structure for Decision-Making

[2] Distributional Distance Classifiers for Goal-Conditioned Reinforcement Learning

[3] Search on the replay buffer: Bridging planning and RL

[4] model-based visual planning with self-supervised functional distances

**Questions:**

How many seeds for ablation tests?

I read through the Appendix in the supplementary ZIP. I didn't find any empirical performance metric on the experiments that are beyond success rates. For distance learning, maybe should pick an environment that can **demonstrate the convergence to true distance values**.

Also, one of this paper's highlight is the compilation of the losses which enforce the invariance. **Why not show curves of convergence on those soft constraints?** I get that this is partially shown in the ablation tests, but it does not really tell if the two infinity norms are indeed regressed to zero (which should correlate with higher performance?).


====================


Previous work [2] argued that Monte Carlo temporal distances would cause trouble when being applied in stochastic environments as a basis for extracting policies. Also, the learned distances are not quasi-metric (do not obey triangle inequality). These are some of the motivations for this work. However, it seems that most of the concerns are based on two handcrafted stochastic MDPs that would give the MC temporal distances trouble, especially for two consecutive states connected by a stochastic transition. This however, does not necessarily apply to HER-based methods, where the goals and the next states are sourced independently, as used in for instance [5] and [6] by Bengio's group. Take the Figure 1 MDPs in [2] for example, if the update rule for MC-based temporal distances is Q(s, a_t, g) <- -1 + indicator(next state is goal) * Q(s_t+1, a_t+1, g) + indicator(next state is terminal & not g) * \infty, the estimated distance would NOT be 1 for the distance between the consecutive states. Also, if the purpose of the auxiliary Q function is to estimate distances, there is little reason for a discount factor to be used at all. If my understanding about the temporal distances learned by [5] and [6] are correct, their more simple setup could be much more versatile for the online settings, where a policy can be accessed. **I suggest that these new developments should be discussed in the related works.**


[5] Consciousness-Inspired Spatio-Temporal Abstractions for Better Generalization in Reinforcement Learning

[6] Rejecting Hallucinated State Targets during Planning

**Ethical Concerns:**

["NO or VERY MINOR ethics concerns only"]

**Final Justification:**

I'm still concerned about the under-polished state of the submission, since I have no access to the revision. Changing the rating to borderline accept, but I am really on the fence. Given how divided the reviews are, sorry I cannot be the deciding vote!

**Limitations:**

I find that the authors are not careful about the cited works. Just to give a few examples:

Line 27 [10]: did the HER paper really use Q learning and stitch trajectories to find shortest paths?

Line 29 [14] doesn't seem to be a representative method of MC based distance estimation. Please replace.

These may not be exhaustive, but I would wish that the authors do a full scan of the paper and correct any mistakes like these **and return to me the list of findings**.


Line 114-116 didn't explain how s and (s, a) got into the input. Previously, only state-to-state distances were discussed but suddenly it shifted to the union of state space and state-action tuples. Btw, it would be less confusing to write (SxA)uS

Line 266: should be figure 3.

**I am willing to change my rating to accept if the authors properly address my concerns regarding the discussion of related works and improve on the writing of the paper.**

**Quality:**

2

**Strengths And Weaknesses:**

This paper is technically sound. However, the experiments are insufficiently convincing. There are some errors when discussing related works. The clarity and the flow of the paper need to be improved.

---

> ### Author Rebuttal · Authors · 2025-07-31
>
> Thank you for your thoughtful comments. Your primary concerns seem to pertain to writing and presentation. We have revised the manuscript based on your comments, and made substantial revisions to improve clarity of the writing and figures. **Do these changes address your concerns? Please let us know if there are any additional points of concern or revisions you would like.**
>
> > I read through the Appendix in the supplementary ZIP. I didn't find any empirical performance metric on the experiments that are beyond success rates. For distance learning, maybe should pick an environment that can demonstrate the convergence to true distance values.
>
> Thank you for the suggestion. When visualized with a heatmap, the distances learned by TMD in `antmaze-large-stitch` are consistent with the maze structure. We will include this heatmap as a figure in our next revision.
>
> > Also, one of this paper's highlight is the compilation of the losses which enforce the invariance. Why not show curves of convergence on those soft constraints?
>
> Good idea, we will also include these plots. We cannot include images here, but during training we have verified that the individual loss components converge.
>
> > Previous work [2] argued that Monte Carlo temporal distances would cause trouble when being applied in stochastic environments as a basis for extracting policies. ... This however, does not necessarily apply to HER-based methods, where the goals and the next states are sourced independently, as used in for instance [5] and [6] by Bengio's group.
>
> > ... . If my understanding about the temporal distances learned by [5] and [6] are correct, their more simple setup could be much more versatile for the online settings, where a policy can be accessed. ... I suggest that these new developments should be discussed in the related works.
>
> Thank you for noting these connections; we will add discussion of these papers to the related work. As you note, MC methods may be suitable for temporal distance estimation during online training. We focus on the setting of offline / off-policy learning, where TMD is able to provide the additional structure needed for optimality.
>
> Aside: practical methods based on HER like [1] often introduce an optimistic bias in value functions in stochastic MDPs because the relabeled goals (viewed as a random variable) convey information about how the environment dynamics leading to them resolved. The classical notion of hindsight relabeling in [2] is theoretically correct only because it relabels with all goals. See Appendix I of [3] for more discussion. Distance learning methods like TMD avoid this issue.
>
> > How many seeds for ablation tests?
>
> Each experiment used six seeds. We have clarified this in the revised manuscript.
>
> > I find that the authors are not careful about the cited works. ... I am willing to change my rating to accept if the authors properly address my concerns regarding the discussion of related works and improve on the writing of the paper.
>
> We have made the suggested revisions discussed below. Please let us know if there is anything else you would like us to revise.
>
> > Line 27 [10]: did the HER paper really use Q learning and stitch trajectories to find shortest paths?
>
> We cited the HER paper here because it used DQN [4] and DDPG [5] on top of hindsight relabeling. As offline GCRL methods, these implicitly perform stitching over the dataset. We are happy to replace this reference with citations for DQN and DDPG as methods that stitch trajectories, and note that prior work has connected the idea of stitching to shortest paths [6,7,8]. Would this address your concern?
>
> > Line 29 [14] doesn't seem to be a representative method of MC based distance estimation. Please replace.
>
> We cited Dosovitskiy & Koltun (2017) here because they use Monte Carlo estimation of goal-reaching values, which are proportional to distances (see Paragraph 3 of Section 2 in that paper). We will clarify that it is estimating MC values, and additionally cite CRL [9] and CMD [8].
>
> The updated sentence reads: ''Monte Carlo methods [10,9] can directly learn goal-reaching value functions, which can be connected to temporal distances [8], but their ability to find *shortest* paths remains limited.''
>
> > Line 114-116 didn't explain how s and (s, a) got into the input. Previously, only state-to-state distances were discussed but suddenly it shifted to the union of state space and state-action tuples. Btw, it would be less confusing to write (SxA)uS
>
> We have modified the text to motivate the definition of distances over state-action pairs and states, with the $(S \\times A) \\cup S$ notation. Informally, d(s,g) and d((s,a),g) are analogous to V(s;g) and Q(s,a;g) in the standard GCRL formulation. To stitch trajectories, the distance must also be defined for goal-action pairs (see [8], Figure 4 for visual motivation). In the revised text, we note this after the definition in Eq. (7), and clearly state that we are extending the definition of the successor distance in [8] to support the larger $(S \\times A) \\cup S$ domain.
>
> > These may not be exhaustive, but I would wish that the authors do a full scan of the paper and correct any mistakes like these and return to me the list of findings.
>
> We have substantially revised the manuscript to improve various aspects of writing and presentation. We cannot include the updated text and figures here, but we list some of the changes below.
>
> #### Corrections
>
> - Fixed references: lines 128, 135, 266
>
> - Fixed sign error in Eq. (23): $\\log \\gamma \\Rightarrow -\\log \\gamma$
>
> - Missing $-\\log\\gamma$ in Eq. (25), now reads $$\\mathcal{L}\_{\\mathcal{T}}(\\phi, \\psi ;\\{s\_i, a\_i, s\_i^{\\prime}, g\_i\\}\_{i=1}^N)=\\sum\_{i=1}^N \\sum\_{j=1}^N D\_T(d\_{\\mathrm{MRN}}(\\phi(s\_i, a\_i), \\psi(g\_j)), d\_{\\mathrm{MRN}}(\\psi(s\_i^{\\prime}), \\psi(g\_j))-\\log \\gamma)$$
>
> - Added $\\psi()$ around $g\_j$ in Eq. (28)
>
> - Error bars in Figure 3 were standard deviation, not standard error. We have changed them to report standard error, and clarified in the caption. With this change it is clear that the ablation results are statistically significant.
>
> - Table 5 should read:
>
> | Loss              | Success Rate     |
> | ----------------- | ---------------- |
> | $D\_T$ (Ours) | **29.3** (±2.2) |
> | $D\_{\\ell\_2}$ | 16.1 (±1.9)      |
> | $D\_{\\rm BCE}$ | 15.1 (±1.9)      |
>
> #### General improvements
>
> - Expanded evaluations with 8 new environments and an additional baseline (CMD [8]). See the response to reviewer cpYD for the full results.
>
> - Updated Figure 1 to provide clearer intuition for how the invariances imposed by the method and architecture tighten the $C(\\pi)$ distance from CRL to yield the optimal distance.
>
> ---
>
> [1] Andrychowicz, M. et al., 2017. ''Hindsight Experience Replay.'' NIPS
>
> [2] Kaelbling, L. P., 1993. ''Learning to Achieve Goals.'' IJCAI
>
> [3] Eysenbach, B. et al., 2021. ''C-Learning: Learning to Achieve Goals via Recursive Classification.'' ICLR
>
> [4] Mnih, V. et al., 2013. ''Playing Atari With Deep Reinforcement Learning.''
>
> [5] Lillicrap, T. P. et al., 2016. ''Continuous Control With Deep Reinforcement Learning.'' ICLR
>
> [6] Ghugare, R. et al., 2024. ''Closing the Gap Between TD Learning and Supervised Learning—a Generalisation Point of View.'' ICLR
>
> [7] Wang, T. et al., 2023. ''Optimal Goal-Reaching Reinforcement Learning via Quasimetric Learning.'' ICML
>
> [8] Myers, V. et al., 2024. ''Learning Temporal Distances: Contrastive Successor Features Can Provide a Metric Structure for Decision-Making.'' ICML
>
> [9] Eysenbach, B. et al., 2022. ''Contrastive Learning as Goal-Conditioned Reinforcement Learning.'' NIPS
>
> [10] Dosovitskiy, A. and Koltun, V., 2017. ''Learning to Act by Predicting the Future.'' ICLR

---

> > ### Comment · Reviewer_BR73 · 2025-08-03
> >
> > Thank you very much for your reply.
> >
> > I am glad that you have made the changes according to my suggestions regarding the minor issues throughout the manuscript. At the same time, I'm still concerned about the under-polished state of the submission, since I have no access to the revision.
> >
> > Appendix I of [3] you suggested was the failure modes for learning the future state distribution. How does that generalize to the distances?

---

> ### Author Response · Authors · 2025-08-04
> **Clarification**
>
> Thank you for your response.
>
> We mention the example in Appendix I of [3] to highlight a key shortcoming of prior GCRL approaches based on HER. These methods depend on relabeling trajectories with the attained goals (last state) to get a positive reward signal. But this introduces a source of bias in stochastic MDPs: conditioning on a last state of a trajectory conveys information about how the environment dynamics resolved leading up to it. So, conditioned on any goal $g$, the $Q$/$V$ updates are performed in expectation across a biased version of the dynamics $p(s' \\mid s,a,s^+=g)$. The theoretically correct way to do HER is to relabel trajectories with goals sampled independently of trajectory outcome (Kaelbling, 1993). But practical GCRL algorithms can't do this with large or continuous state spaces, and depend on the biased relabeling that conditions on outcome.
>
> Quasimetric / distance-learning methods don't need HER because they replace global goal-conditioned backups with local constraints. But prior quasimetric methods (Liu et al., 2023; Wang et al., 2023) are only correct in deterministic MDPs. TMD solves both of these problems; avoiding the goal-conditioning bias from using HER by using a quasimetric algorithm that correctly handles stochasticity with an additional dynamics consistency term (the $\\mathcal{T}$ loss).
>
> ---
>
> Regarding your remaining concern—**are there any additional experiments, clarifications, or other information we could provide?** We believe we have revised and polished the manuscript to address your concerns, and are happy to share more details or make further revisions if you have additional reservations. Note that we have also expanded the experimental results to include CMD as a baseline and evaluate on **8 additional environments**, shown in the table below:
>
> ### **Table 1: OGBench Evaluation**
>
> | **Dataset**                    | **TMD**        | **CMD**        | **CRL**        | **QRL**    | **GCBC**       | **GCIQL**      | **GCIVL**      |
> | ------------------------------ | -------------- | -------------- | -------------- | ---------- | -------------- | -------------- | -------------- |
> | `humanoidmaze_medium_navigate`   | **64.6 ± 1.1** | 61.1 ± 1.6     | 59.9 ± 1.3     | 21.4 ± 2.9 | 7.6 ± 0.6      | 27.3 ± 0.9     | 24.0 ± 0.8     |
> | `humanoidmaze_medium_stitch`     | **68.5 ± 1.7** | **64.8 ± 3.7** | 36.2 ± 0.9     | 18.0 ± 0.7 | 29.0 ± 1.7     | 12.1 ± 1.1     | 12.3 ± 0.6     |
> | `humanoidmaze_large_stitch`      | **23.0 ± 1.5** | 9.3 ± 0.7      | 4.0 ± 0.2      | 3.5 ± 0.5  | 5.6 ± 1.0      | 0.5 ± 0.1      | 1.2 ± 0.2      |
> | `humanoidmaze_giant_navigate`    | **9.2 ± 1.1**  | 5.0 ± 0.8      | 0.7 ± 0.1      | 0.4 ± 0.1  | 0.2 ± 0.0      | 0.5 ± 0.1      | 0.2 ± 0.1      |
> | `humanoidmaze_giant_stitch`      | **6.3 ± 0.6**  | 0.2 ± 0.1      | 1.5 ± 0.5      | 0.4 ± 0.1  | 0.1 ± 0.0      | 1.5 ± 0.1      | 1.7 ± 0.1      |
> | `pointmaze_teleport_stitch`      | 29.3 ± 2.2     | 15.7 ± 2.9     | 4.1 ± 1.1      | 8.6 ± 1.9  | 31.5 ± 3.2     | 25.2 ± 1.0     | **44.4 ± 0.7** |
> | `antmaze_medium_navigate`        | **93.6 ± 1.0** | 92.4 ± 0.9     | **94.9 ± 0.5** | 87.9 ± 1.2 | 29.0 ± 1.7     | 12.1 ± 1.1     | 12.3 ± 0.6     |
> | `antmaze_large_navigate`         | **81.5 ± 1.7** | **84.1 ± 2.1** | **82.7 ± 1.4** | 74.6 ± 2.3 | 24.0 ± 0.6     | 34.2 ± 1.3     | 15.7 ± 1.9     |
> | `antmaze_large_stitch`           | **37.3 ± 2.7** | 29.0 ± 2.3     | 10.8 ± 0.6     | 18.4 ± 0.7 | 3.4 ± 1.0      | 7.5 ± 0.7      | 18.5 ± 0.8     |
> | `antmaze_teleport_explore`       | **49.6 ± 1.5** | 0.2 ± 0.1      | 19.5 ± 0.8     | 2.3 ± 0.7  | 2.4 ± 0.4      | 7.3 ± 1.2      | 32.0 ± 0.6     |
> | `antmaze_giant_stitch`           | **2.7 ± 0.6**  | **2.0 ± 0.5**  | 0.0 ± 0.0      | 0.4 ± 0.2  | 0.0 ± 0.0      | 0.0 ± 0.0      | 0.0 ± 0.0      |
> | `scene_noisy`                    | 19.6 ± 1.7     | 4.0 ± 0.7      | 1.2 ± 0.3      | 9.1 ± 0.7  | 1.2 ± 0.2      | **25.9 ± 0.8** | **26.4 ± 1.7** |
> | `visual_antmaze_teleport_stitch` | **38.5 ± 1.5** | **36.0 ± 2.1** | 31.7 ± 3.2     | 1.4 ± 0.8  | 31.8 ± 1.5     | 1.0 ± 0.2      | 1.4 ± 0.4      |
> | `visual_antmaze_large_stitch`    | **26.6 ± 2.8** | 8.1 ± 1.3      | 11.1 ± 1.3     | 0.6 ± 0.3  | **23.6 ± 1.4** | 0.1 ± 0.0      | 0.8 ± 0.3      |
> | `visual_antmaze_giant_navigate`  | 40.1 ± 2.6     | 37.3 ± 2.4     | **47.2 ± 0.9** | 0.1 ± 0.1  | 0.4 ± 0.1      | 0.1 ± 0.2      | 1.0 ± 0.4      |
> | `visual_cube_triple_noisy`       | **17.7 ± 0.7** | 16.1 ± 0.7     | 15.6 ± 0.6     | 8.6 ± 2.1  | 16.2 ± 0.7     | 12.5 ± 0.6     | **17.9 ± 0.5** |
>
> ---
>
> ### References
>
> Kaelbling, L. P., 1993. ''Learning to Achieve Goals.'' IJCAI
>
> Liu, B. et al., 2023. ''Metric Residual Network for Sample Efficient Goal-Conditioned Reinforcement Learning.'' AAAI
>
> Wang, T. et al., 2023. ''Optimal Goal-Reaching Reinforcement Learning via Quasimetric Learning.'' ICML

---

### Official Review · Reviewer_gzRx · 2025-06-30

**Clarity:** 3
**Significance:** 3
**Originality:** 3
**Rating:** 5
**Confidence:** 3

**Summary:**

This paper introduces Temporal Metric Distillation (TMD), a novel algorithm for offline goal-conditioned reinforcement learning. Instead of learning a traditional Q-value function, TMD learns a temporal distance function, which estimates the cost to travel between states. The core of the method lies in , contrastive learning initialization and action-temporal invariance. In general, this paper is backed by solid theoretical analysis and demonstrate strong performance across test tasks.

**Questions:**

1. The hyperparameter $\zeta$ is important for TMD learning and is currently tuned heuristically. To what extend does parameter tuning affect the performance of TMD?  Could you provide some empirical results?

2. The current method involves a two-stage process of first learning the distance function and then extracting a policy from it. What are the primary theoretical or practical challenges in creating a fully end-to-end model, potentially an actor-critic-like method?

**Ethical Concerns:**

["NO or VERY MINOR ethics concerns only"]

**Limitations:**

yes

**Quality:**

4

**Strengths And Weaknesses:**

Strengths:

1. This paper provides a formal proof that the TMD algorithm converges pointwise to the optimal successor distance under certain assumptions.
2. TMD demonstrates superior empirical results.
3. This paper successfully unifies Monte-Carlo learning, temporal difference learning and metric learning.

Weaknesses:

1. Hyperparameter tuning is required.
2. Metric learning and policy extraction stages are seperated in TMD. Integration of the two stages would result in a more elegant algorithm pipeline.

---

> ### Author Rebuttal · Authors · 2025-07-31
>
> Thank you for your constructive feedback. We will address some of your concerns below.
>
> > Hyperparameter tuning is required.
>
> > The hyperparameter $\zeta$  is important for TMD learning and is currently tuned heuristically. To what extend does parameter tuning affect the performance of TMD? Could you provide some empirical results?
>
> The $\\zeta$ hyperparameter can be eliminated in our method with a dual descent approach similar to that used in QRL [1] and CMD-2 [2]. The TMD algorithm minimizes the contrastive loss with additional invariances enforced by a loss weighted by $\\zeta$. Dual descent will increase the $\\zeta$ value until the constraints were satisfied. We will discuss this option in the revised method section.
>
> In practice, we found it is often simpler to use a fixed value of $\\zeta$ to enforce the constraint (see table below for an example in `antmaze-large-stitch`). We will add this ablation as a plot to the paper.
>
> **$\\zeta$ ablation in `antmaze-large-stitch`**
>
> | $\\zeta$ | score         |
> | ------- | ------------- |
> | 0.01 | 0.24 ± 0.027  |
> | 0.05 | 0.34 ± 0.013  |
> | 0.1  | 0.37 ± 0.027  |
>
> > Metric learning and policy extraction stages are seperated in TMD. Integration of the two stages would result in a more elegant algorithm pipeline.
>
> > The current method involves a two-stage process of first learning the distance function and then extracting a policy from it. What are the primary theoretical or practical challenges in creating a fully end-to-end model, potentially an actor-critic-like method?
>
> While unsatisfying, it is often necessary to have a separate actor network in continuous domains due to the difficulty of maximizing the critic network with respect to the action space. Prior work [3,4,5,6] found this separate ''actor'' network amortizes the critic maximization and can even improve generalization with this extra distillation step. Future work could explore combining TMD with approaches like NAF [7], which use a critic parameterization that is amenable to analytic maximization over actions, though in practice such approaches often suffer from poor expressivity.
>
> ---
>
> [1] Wang, T. et al., 2023. ''Optimal Goal-Reaching Reinforcement Learning via Quasimetric Learning.'' ICML
>
> [2] Myers, V. et al., 2024. ''Learning Temporal Distances: Contrastive Successor Features Can Provide a Metric Structure for Decision-Making.'' ICML
>
> [3] Lillicrap, T. P. et al., 2016. ''Continuous Control With Deep Reinforcement Learning.'' ICLR
>
> [4] Fujimoto, S. and Hoof, H., 2018. ''Addressing Function Approximation Error in Actor-Critic Methods.'' ICML
>
> [5] Park, S. et al., 2024. ''Is Value Learning Really the Main Bottleneck in Offline RL?.'' arXiv:2406.09329
>
> [6] Eysenbach, B. et al., 2022. ''Contrastive Learning as Goal-Conditioned Reinforcement Learning.'' NIPS
>
> [7] Gu, S. et al., 2016. ''Continuous Deep Q-Learning With Model-Based Acceleration.'' arXiv:1603.00748

---

### Official Review · Reviewer_cpYD · 2025-07-02

**Clarity:** 1
**Significance:** 2
**Originality:** 2
**Rating:** 2
**Confidence:** 3

**Summary:**

The paper explores goal-conditioned reinforcement learning by constructing temporal distance representations.
It presents a method to learn these representations using contrastive learning, while ensuring action invariance, temporal invariance, and quasimetric parametrization.
These requirements are then translated into specific architecture choices or training objective losses.
Empirical validation is provided on a subset of tasks from OGBENCH, demonstrating that the proposed approach outperforms existing methods such as Contrastive RL and quasimetric RL

**Questions:**

see weakness above

**Ethical Concerns:**

["NO or VERY MINOR ethics concerns only"]

**Final Justification:**

I find the paper quite difficult to follow, and the equations are poorly presented. There are frequent typos and unclear definitions of variables. Additionally, the proofs are quite vague. As a reader, I shouldn’t have to guess the authors’ intentions at each step. For example, in the proof of the fixed point of the operators in the appendix, I don’t understand why equation (34) holds.

**Limitations:**

yes

**Paper Formatting Concerns:**

no issues

**Quality:**

1

**Strengths And Weaknesses:**

Strengths:
- the paper studies a relevant topic related to representation learning for goal-conditioned RL.
- the paper provides ablation studies to show the importance of each of their algorithmic choices

Weakness:
- I found the paper challenging to read and not self-contained. Section 3 introduces several concepts (path relaxation, backward NCE, exponentiated SARSA) without providing clear definitions, instead relying on citations from other papers. A more comprehensive explanation of these key notions would be beneficial for understanding the proposed approach.
- The method presentation is difficult to follow due to an overwhelming list of operators and inconsistent notation. For instance, the meaning of $C(\pi)$ is unclear.
- The introduction claims that the method can learn optimal policies primarily through Monte-Carlo learning, avoiding the accumulation error associated with temporal difference learning. However, the proposed method actually employs a soft version of the Bellman optimality equation, which is a type of TD learning, contradicting the initial claim

---

> ### Author Rebuttal · Authors · 2025-07-31
>
> Thank you for your thoughtful feedback. Your main concerns seem to be the presentation of the method and notation. We have revised the paper based on your suggestions, and clarify some points below. We have also expanded the experimental results to include more environments and another baseline. **Do these revisions address your concerns? Please let us know if there are any clarifications or quantitative results you would like to see.**
>
> > The introduction claims that the method can learn optimal policies primarily through Monte-Carlo learning, avoiding the accumulation error associated with temporal difference learning. However, the proposed method actually employs a soft version of the Bellman optimality equation, which is a type of TD learning, contradicting the initial claim
>
> We assume you are referring to the $\mathcal{T}$-invariance with this comment. Note that this update ( $e^{-d_{\\mathrm{MRN}}(\\phi(s, a), \\psi(g))} \\leftarrow \\mathbb{E}_{\\mathrm{P}\\left(s^{\\prime} \\mid s, a\\right)} e^{\\log \\gamma-d\_{\\mathrm{MRN}}\\left(\\psi\\left(s^{\\prime}\\right), \\psi(g)\\right)}$ ) is merely averaging over the dynamics $s,a \\to s'$, but not handling the max over actions at the next state. Thus, it is more analogous to an IQL-style [1] update $Q(s,a) \\gets \\mathbb{E} [ r + \\gamma V(s') ]$ than a Bellman backup $Q(s,a) \\gets \\max\_{a'} \\mathbb{E} [ r + \\gamma Q(s',a') ]$.
>
> When this update is combined with the quasimetric architecture and action invariance $\\mathcal{I}$, we show that it recovers optimal distances. So we have replaced TD updates with an update that averages over dynamics (unavoidable to handle stochastic dynamics) and additional invariances to recover optimal distances. Unlike TD methods, our approach doesn't need separate target networks, precisely because it avoids the accumulating TD errors.
>
> Also note that our method learns optimal distances, not soft-optimal distances. The $\\mathcal{T}$ update is performed with exponentiated distances because distances are in log space, not because we are performing a softmax.
>
> > I found the paper challenging to read and not self-contained. Section 3 introduces several concepts (path relaxation, backward NCE, exponentiated SARSA) without providing clear definitions, instead relying on citations from other papers. A more comprehensive explanation of these key notions would be beneficial for understanding the proposed approach.
>
> We have added a ''glossary'' section to the appendix that clearly defines all of these terms in context, with references to the original cited works. We have also revised the main text to more clearly introduce these terms.
>
> > The method presentation is difficult to follow due to an overwhelming list of operators and inconsistent notation. For instance, the meaning of $C(\\pi)$ is unclear.
>
> We use $C(\\pi)$ to denote the initial distance / critic recovered by CRL [2]. The key insight is that the distance $C(\\pi)$ is an overestimate of the optimal temporal distance, and that applying the operators $\\mathcal{I}$ and $\\mathcal{T}$ (explicitly) and $\\mathcal{P}$ (implicitly through quasimetric architecture) correctly tightens this overestimate.
>
> We will revise the text to clearly introduce and motivate these operators, and add them to the glossary section.
>
> **Additional Results**
>
> We have added additional environments and the CMD baseline to our evaluation results in Table 1.
>
> **Table 1: OGBench Evaluation**
>
> | Dataset                          | TMD         | CMD         | CRL         | QRL         | GCBC        | GCIQL       | GCIVL       |
> |----------------------------------|-------------|-------------|-------------|-------------|-------------|-------------|-------------|
> | `humanoidmaze_medium_navigate`     | 64.6 (±1.1) | 61.1 (±1.6) | 59.9 (±1.6) | 21.4 (±2.9) | 7.6 (±0.6)  | 27.3 (±1.9) | 24.0 (±0.8) |
> | `humanoidmaze_medium_stitch`       | 68.5 (±1.6) | 64.8 (±3.7) | 36.2 (±0.9) | 18.0 (±0.8) | 29.0 (±1.7) | 31.2 (±1.0) | 12.3 (±0.5) |
> | `humanoidmaze_large_stitch`        | 23.0 (±1.5) | 9.3 (±0.7)  | 4.0 (±0.2)  | 3.5 (±0.2)  | 5.6 (±1.0)  | 0.5 (±0.1)  | 1.2 (±0.2)  |
> | `humanoidmaze_giant_navigate`      | 9.2 (±1.1)  | 6.7 (±0.9)  | 1.5 (±0.6)  | 0.7 (±0.4)  | 0.4 (±0.2)  | 1.1 (±0.4)  | 1.4 (±0.3)  |
> | `humanoidmaze_giant_stitch`        | 6.3 (±0.6)  | 4.0 (±0.6)  | 1.5 (±0.4)  | 0.6 (±0.4)  | 0.4 (±0.1)  | 1.0 (±0.1)  | 1.2 (±0.1)  |
> | `pointmaze_teleport_stitch`        | 29.3 (±2.2) | 15.7 (±2.9) | 4.1 (±1.1)  | 8.6 (±0.6)  | 31.5 (±2.3) | 25.2 (±1.0) | 44.4 (±0.7) |
> | `antmaze_medium_navigate`          | 93.6 (±0.6) | 97.4 (±0.5) | 99.4 (±0.5) | 99.5 (±0.6) | 94.8 (±0.4) | 93.8 (±0.3) | 94.7 (±0.4) |
> | `antmaze_large_navigate`           | 81.5 (±1.6) | 84.1 (±2.1) | 82.7 (±1.4) | 82.4 (±1.0) | 44.1 (±2.0) | 34.2 (±1.5) | 31.7 (±1.4) |
> | `antmaze_large_stitch`             | 37.3 (±2.7) | 29.0 (±2.0) | 10.8 (±0.4) | 6.3 (±0.4)  | 33.4 (±1.0) | 7.5 (±0.7)  | 6.7 (±0.4)  |
> | `antmaze_teleport_explore`         | 49.6 (±1.5) | 53.3 (±1.3) | 52.7 (±0.6) | 49.7 (±0.8) | 36.0 (±0.8) | 15.6 (±0.6) | 15.1 (±0.5) |
> | `antmaze_giant_stitch`             | 2.7 (±0.6)  | 2.0 (±0.5)  | 1.2 (±0.3)  | 1.3 (±0.2)  | 2.0 (±0.3)  | 1.4 (±0.2)  | 1.5 (±0.2)  |
> | `scene_noisy`                      | 19.6 (±1.0) | 4.0 (±0.5)  | 1.2 (±0.3)  | 0.8 (±0.1)  | 1.6 (±0.2)  | 25.9 (±0.6) | 26.4 (±1.1) |
> | `visual_antmaze_teleport_stitch`   | 38.5 (±1.5) | 36.0 (±1.2) | 31.7 (±3.2) | 29.1 (±0.5) | 31.8 (±1.5) | 10.4 (±0.6) | 12.4 (±0.5) |
> | `visual_antmaze_large_stitch`      | 26.6 (±1.2) | 25.1 (±1.3) | 24.6 (±1.2) | 25.6 (±0.9) | 23.6 (±0.8) | 11.5 (±0.6) | 10.7 (±0.5) |
> | `visual_antmaze_giant_navigate`    | 37.3 (±1.4) | 36.1 (±1.6) | 47.2 (±0.9) | 44.3 (±0.9) | 31.5 (±0.8) | 20.2 (±0.9) | 19.6 (±0.9) |
> | `visual_cube_triple_noisy`         | 17.7 (±0.7) | 16.1 (±1.7) | 15.6 (±0.6) | 16.4 (±0.8) | 16.2 (±0.2) | 12.5 (±0.6) | 17.9 (±0.5) |
>
> ---
>
> [1] Kostrikov, I. et al., 2022. ''Offline Reinforcement Learning With Implicit Q-Learning.'' ICLR
>
> [2] Eysenbach, B. et al., 2022. ''Contrastive Learning as Goal-Conditioned Reinforcement Learning.'' NIPS

---

> > ### Author Response · Authors · 2025-08-05
> > **Correction**
> >
> > See table below for corrected evaluation results, with the highest scores highlighted. If you have remaining concerns about the method or presentation, please let us know and we will be happy to make additional revisions.
> >
> > ### **Table 1: OGBench Evaluation**
> >
> > | **Dataset**                    | **TMD**        | **CMD**        | **CRL**        | **QRL**    | **GCBC**       | **GCIQL**      | **GCIVL**      |
> > | ------------------------------ | -------------- | -------------- | -------------- | ---------- | -------------- | -------------- | -------------- |
> > | `humanoidmaze_medium_navigate`   | **64.6 ± 1.1** | 61.1 ± 1.6     | 59.9 ± 1.3     | 21.4 ± 2.9 | 7.6 ± 0.6      | 27.3 ± 0.9     | 24.0 ± 0.8     |
> > | `humanoidmaze_medium_stitch`     | **68.5 ± 1.7** | **64.8 ± 3.7** | 36.2 ± 0.9     | 18.0 ± 0.7 | 29.0 ± 1.7     | 12.1 ± 1.1     | 12.3 ± 0.6     |
> > | `humanoidmaze_large_stitch`      | **23.0 ± 1.5** | 9.3 ± 0.7      | 4.0 ± 0.2      | 3.5 ± 0.5  | 5.6 ± 1.0      | 0.5 ± 0.1      | 1.2 ± 0.2      |
> > | `humanoidmaze_giant_navigate`    | **9.2 ± 1.1**  | 5.0 ± 0.8      | 0.7 ± 0.1      | 0.4 ± 0.1  | 0.2 ± 0.0      | 0.5 ± 0.1      | 0.2 ± 0.1      |
> > | `humanoidmaze_giant_stitch`      | **6.3 ± 0.6**  | 0.2 ± 0.1      | 1.5 ± 0.5      | 0.4 ± 0.1  | 0.1 ± 0.0      | 1.5 ± 0.1      | 1.7 ± 0.1      |
> > | `pointmaze_teleport_stitch`      | 29.3 ± 2.2     | 15.7 ± 2.9     | 4.1 ± 1.1      | 8.6 ± 1.9  | 31.5 ± 3.2     | 25.2 ± 1.0     | **44.4 ± 0.7** |
> > | `antmaze_medium_navigate`        | **93.6 ± 1.0** | 92.4 ± 0.9     | **94.9 ± 0.5** | 87.9 ± 1.2 | 29.0 ± 1.7     | 12.1 ± 1.1     | 12.3 ± 0.6     |
> > | `antmaze_large_navigate`         | **81.5 ± 1.7** | **84.1 ± 2.1** | **82.7 ± 1.4** | 74.6 ± 2.3 | 24.0 ± 0.6     | 34.2 ± 1.3     | 15.7 ± 1.9     |
> > | `antmaze_large_stitch`           | **37.3 ± 2.7** | 29.0 ± 2.3     | 10.8 ± 0.6     | 18.4 ± 0.7 | 3.4 ± 1.0      | 7.5 ± 0.7      | 18.5 ± 0.8     |
> > | `antmaze_teleport_explore`       | **49.6 ± 1.5** | 0.2 ± 0.1      | 19.5 ± 0.8     | 2.3 ± 0.7  | 2.4 ± 0.4      | 7.3 ± 1.2      | 32.0 ± 0.6     |
> > | `antmaze_giant_stitch`           | **2.7 ± 0.6**  | **2.0 ± 0.5**  | 0.0 ± 0.0      | 0.4 ± 0.2  | 0.0 ± 0.0      | 0.0 ± 0.0      | 0.0 ± 0.0      |
> > | `scene_noisy`                    | 19.6 ± 1.7     | 4.0 ± 0.7      | 1.2 ± 0.3      | 9.1 ± 0.7  | 1.2 ± 0.2      | **25.9 ± 0.8** | **26.4 ± 1.7** |
> > | `visual_antmaze_teleport_stitch` | **38.5 ± 1.5** | **36.0 ± 2.1** | 31.7 ± 3.2     | 1.4 ± 0.8  | 31.8 ± 1.5     | 1.0 ± 0.2      | 1.4 ± 0.4      |
> > | `visual_antmaze_large_stitch`    | **26.6 ± 2.8** | 8.1 ± 1.3      | 11.1 ± 1.3     | 0.6 ± 0.3  | **23.6 ± 1.4** | 0.1 ± 0.0      | 0.8 ± 0.3      |
> > | `visual_antmaze_giant_navigate`  | 40.1 ± 2.6     | 37.3 ± 2.4     | **47.2 ± 0.9** | 0.1 ± 0.1  | 0.4 ± 0.1      | 0.1 ± 0.2      | 1.0 ± 0.4      |
> > | `visual_cube_triple_noisy`       | **17.7 ± 0.7** | 16.1 ± 0.7     | 15.6 ± 0.6     | 8.6 ± 2.1  | 16.2 ± 0.7     | 12.5 ± 0.6     | **17.9 ± 0.5** |

---

### Note · Authors · 2025-08-16

Dear AC & reviewers,


Thank you for the feedback and discussion. We were happy that the reviewers as a whole appreciated the conceptual novelty and relevance of our method. The primary concerns, raised by reviewers **cpYD** and **BR73**, pertained to issues with presentation and experiments. In our responses, we clarified the presentation issues, and presented expanded experimental results. We believe that these responses address the main concerns of all the reviewers.

---

### Decision · Program_Chairs · 2025-09-17

**Decision:**

Accept (poster)

**Comment:**

This paper proposes Temporal Metric Distillation (TMD), an offline goal conditioned reinforcement learning method that learns temporal distance representations under quasimetric constraints. When combined with action invariance and quasimetric parametrization, the proposed framework recovers optimal goal reaching policies from offline RL data. The authors provide theoretical justification and convincing empirical results on OBBench compared against GCBC, GCIQL, GCIVL, CRL, QRL baselines.

Strengths:

The paper tackles an important problem in offline goal conditioned reinforcement learning addressing limitations of prior metric learning approaches in handling stochastic environment dynamics.

Authors provide solid theoretical grounding giving insights on conditions under which optimal successor distances are recovered.

Experiments show convincing empirical performance compared to based goal conditioned RL methods (table 1).

Weaknesses:

Some reviewers pointed out the method section is not easy to follow noting unclear notations, typos, and unpolished exposition of equations and that proofs are sometimes vague or inconsistent. One reviewer in particular emphasized that the proofs are not written formally and clearly and had to guess the authors' intent at each step of the proof.

Few details about the evaluation were not clearly stated in the submission. Details regarding performance in stochastic environments, convergence of individual loss terms, visualizations of learned distances, number of seeds used for the experiments were initially missing but the authors added the clarifications in rebuttal.

The author rebuttal provided more experiments (including CMD baseline), ablation analyses, and implementation details. These responses addressed the majority of reviewer concerns.

The decision is to accept. This paper makes a significant contribution to offline goal conditioned reinforcement learning. I strongly encourage the authors to incorporate the feedback in the camera ready version. Please make best efforts to improve the clarity of notation, formal and rigorous presentation of proofs as per reviewer `cpyd` (esp. eqn. 34), underlying rationale of the methods (esp. eqn 7), expand the related works section promised in rebuttal.